



# Radiation in fog: Quantification of the impact on fog liquid water based on ground-based remote sensing

Eivind G. Wærsted[1], Martial Haeffelin[2], Jean-Charles Dupont[3], Julien Delanoë[4], Philippe Dubuisson[5]

[1]Laboratoire de Météorologie Dynamique, École Polytechnique, Université Paris-Saclay, 91128 Palaiseau, France
[2]Institut Pierre Simon Laplace, École Polytechnique, CNRS, Université Paris-Saclay, 91128 Palaiseau, France
[3]Institut Pierre-Simon Laplace, École Polytechnique, UVSQ, Université Paris-Saclay, 91128 Palaiseau, France
[4]Laboratoire Atmosphères, Milieux, Observations Spatiales/UVSQ/CNRS/UPMC, 78280 Guyancourt, France
[5]Laboratoire d'Optique Atmosphérique, Univ. Lille – UMR CNRS 8518, F -59000 Lille, France

*Correspondence to*: Eivind G. Wærsted (ewaersted@lmd.polytechnique.fr)

**Abstract.** Radiative cooling and heating impact the liquid water balance of fogs and therefore play an important role in determining their persistence or dissipation. We demonstrate that a quantitative analysis of the radiation-driven condensation and evaporation is possible in real-time using ground-based remote sensing observations (cloud radar, ceilometer, microwave radiometer). Seven continental fog events in mid-latitude winter are studied. The longwave (LW) radiative cooling of the fog is able to produce 40–70 g m$^{-2}$ h$^{-1}$ of liquid water by condensation when the fog liquid water path exceeds
30 g m$^{-2}$ and there are no clouds above the fog, which corresponds to renewing the fog water in 1–2 hours. The variability is related to fog temperature and atmospheric humidity, with warmer fogs below drier atmospheres producing more liquid water. The appearance of a cloud layer above the fog strongly reduces this cooling, especially a low cloud (up to 100 %), thereby perturbing the liquid water balance in the fog, and may therefore induce fog dissipation. Shortwave (SW) radiative heating by absorption by fog droplets is smaller than the LW cooling, but it can contribute significantly, inducing 10–15 g
m$^{-2}$ h$^{-1}$ of evaporation in thick fogs at (winter) midday. We also find that the absorption of SW radiation by aerosols in the fog may strongly increase this evaporation rate if a large concentration of absorbing aerosols is present, but that this increase likely is below 30 % in most cases. The absorbed radiation at the surface can reach 40–120 W m$^{-2}$ during daytime depending on the fog thickness. As in situ measurements indicate that 20–40 % of this energy is transferred to the fog as sensible heat, this surface absorption can contribute importantly to heating and evaporation of the fog, up to 30 g m$^{-2}$ h$^{-1}$ for thin fogs.

# 1 Introduction

Fog is defined as the presence of droplets in the vicinity of the Earth's surface reducing the visibility to below 1 km (American Meteorological Society, 2017). Reduced visibility associated with fog is a major concern for traffic safety, in particular on airports, where delays caused by low visibility procedures causes significant financial losses (Gultepe et al., 2009). In spite of significant advances in the skills of numerical weather forecast models in recent decades, the timing of the
appearance and dissipation of fog is poorly forecasted (Bergot et al., 2007; Steeneveld et al., 2015). Fog is difficult to model



with numerical weather forecast models because of its local nature and the subtle balance between the physical processes that govern its life cycle, which must be parametrized in the models (Steeneveld et al., 2015). Detailed ground-based observations of a fog situation in real-time therefore have a potential for capturing information which is missed by the models and which could help estimate whether the fog will dissipate or persist in the near future.

5       Continental fogs often form by radiative cooling of the surface under clear skies (radiation fog) or by the lowering of the base of a pre-existing low stratus cloud to ground level (Gultepe et al., 2007; Haeffelin et al., 2010). Once the fog has formed, its evolution depends on the physical processes that impact the liquid water. A delicate balance between radiative cooling, turbulent mixing and droplet sedimentation has been found in observational and modelling studies of radiation fog (Brown and Roach, 1976; Zhou and Ferrier, 2008; Price et al., 2015). While radiative cooling produces liquid water by

supersaturation, turbulent mixing usually is a loss mechanism for liquid water through the mixing of the fog with drier air or turbulent deposition of liquid water on the surface (Gultepe et al., 2007).

      Three radiative processes affect the evolution of the fog by cooling or heating it. Firstly, the cooling from emission of thermal (longwave, LW) radiation at the fog top produces liquid water by condensation, which maintains the fog against the processes that deplete the liquid water. The advection of a cloud layer above an existing fog will shelter the fog from this

radiative cooling and can therefore be an efficient dissipation mechanism (Brown and Roach, 1976). Secondly, solar (shortwave, SW) radiation will be absorbed by the fog droplets, mainly in the near-infrared spectrum (Ackerman and Stephens, 1987), which causes heating and subsequent evaporation and loss of liquid water. Finally, heating of the ground by absorption of SW radiation can cause a sensible heat transfer to the fog, causing the fog to evaporate from below (Brown and Roach, 1976). Fog therefore often forms during the night, when thermal cooling dominates, and dissipates a few hours after

sunrise due to the increasing heating from solar radiation (Tardif and Rasmussen, 2007; Haeffelin et al., 2010).

      In this study, we aim to quantify the impact of the three radiative processes mentioned above on the liquid water of continental fogs, based on continuous observations of the atmospheric column obtained from ground-based remote sensing instruments. Such instruments have been proven useful for the study of fog life cycle: the attenuated backscatter from a ceilometer can detect the growth of aerosols preceding fog formation (Haeffelin et al., 2016), while a cloud radar can provide

information about the fog vertical development and properties once it has formed (Teshiba et al., 2004; Boers et al., 2012; Dupont et al., 2012). We search answers to the following questions: How large is the rate of condensation or evaporation induced by each of the three radiative processes? How much does this vary from one situation to another, and which atmospheric parameters are responsible for this variability? How can the magnitude of these impacts be quantified using ground-based remote sensing, and how large are the uncertainties?

30       In Sect. 2, we define the quantitative parameters used to describe the three radiative processes and how they are calculated, and we present the instruments, the radiative transfer code and the fog events studied. Section 3 provides a detailed description of how the observations are used to provide input to the radiative transfer code. In Sect. 4, we present the results when applying the methodology to the observed fog events. In Sect. 5, we discuss the uncertainties of the



methodology and explore how sensitive the radiative processes are to different aspects of the atmospheric conditions. We also discuss the implications of our findings for the dissipation of fogs. Finally, our conclusions are given in Sect. 6.

## 2 Data and method

### 2.1 Overview of the approach

Each of the three radiative processes in the fog is studied using a quantitative parameter. For the process of cooling due to LW emission, we calculate the rate of condensation in the whole fog (in $g\ m^{-2}\ h^{-1}$) that would occur due to this radiative cooling if no other processes occurred, and we call it $C_{LW}$ for short. Similarly, we calculate the evaporation rate due to SW heating inside the fog (in $g\ m^{-2}\ h^{-1}$) and call it $E_{SW}$. The third process is the radiative heating of the surface, which will stimulate a sensible heat flux from the surface to the overlying fog when the surface becomes warmer than the fog. With this

process in mind, our third parameter is the net radiative flux (SW+LW) absorbed at the surface (in $W\ m^{-2}$), $R_{net,s}$ for short. The relationship between $R_{net,s}$ and the sensible heat flux is also studied (Sect. 4.2).

Figure 1 shows schematically how the three parameters are calculated. Measurements from several in situ and remote sensing instruments (presented in Sect. 2.2) are used to estimate the input data to a radiative transfer model (presented in Sect. 2.3). The input data involve vertical profiles of clouds, temperature and humidity. The details of how we

go from measurements to input data are presented in Sect. 3. The radiative transfer model calculates the profile of radiative fluxes and heating rates. The computed fluxes can be compared to measured fluxes at 10 m for validation. From the radiative heating rates, we can calculate the rates of condensation or evaporation in $g\ m^{-2}\ h^{-1}$ (explained in Sect. 2.4).

### 2.2 Observational site and instrumentation

The multi-instrumental atmospheric observatory SIRTA in Palaiseau, 20 km south of Paris (France), provides routine

measurements of a large number of meteorological variables since 2002 (Haeffelin et al., 2005). In situ and remote sensing observations taken at this site have been used to study fog life cycle since 2006 in the framework of the ParisFog project (Haeffelin et al., 2010). In this study, we use the observations from several instruments of SIRTA (Table 1) to analyse periods when fog occurred. The observatory is located in a suburban area, with surroundings characterized by small-scale heterogeneities including an open field, a lake and a small wood.

In situ measurements of visibility, air temperature, wind speed, surface skin temperature and SW and LW radiative fluxes are continuously taken in the surface layer at the observatory. Radiosondes measuring the temperature and humidity profiles between ground level and 30 km are launched twice a day from the Météo-France Trappes station, located 15 km west of SIRTA. Measurements of sensible heat flux taken at 2 m using the eddy correlation method based on CSAT-3 sonic anemometer are applied to study the relationship between surface radiation budget and surface sensible heat flux.

A Vaisala CL31 ceilometer operating at 905 nm provides the profile of (attenuated) light backscatter at 15 m vertical resolution, from which the cloud base height can be determined.


The 95 GHz cloud radar BASTA is a newly developed cloud radar, whose first prototype has been successfully operating at SIRTA since 2010 (Delanoë et al., 2016), observing the vertical profile of clouds in zenith direction. Unlike traditional radars, which emit short, powerful pulses of radiation, BASTA instead uses the frequency-modulated continuous wave technique, which makes it much less expensive than traditional radars (Delanoë et al., (2016), http://basta.projet.latmos.ipsl.fr/). Unlike the ceilometer pulse, the signal of the radar is only weakly attenuated by clouds and can therefore observe thick and multilevel cloud layers. However, the signal weakens with the distance to the target, which limits the ability of the radar to detect clouds with small droplets. BASTA therefore operates at four different modes, with vertical resolution of 12.5 m, 25 m, 100 m and 200 m, respectively. The radar switches systematically between the four modes so that each of them produces a measurement every 12 seconds based on 3 seconds of integration time. Better vertical resolution comes at the cost of sensitivity. The BASTA prototype used in this study can detect clouds at 1 km range (i.e. altitude) with reflectivity (see Sect. 3.2) above -27.5, -32, -38 and -41 dBZ with the 12.5m, 25m, 100m and 200m mode, respectively. This lower limit for detection increases approximately with the square of the range, i.e. with 6 dBZ when the range increases by a factor of two. However, a new prototype that recently has been developed has improved the sensitivity with about 12 dBZ relative to the first prototype on all levels. The lowest ≈3 altitude levels in the radar data cannot be used because of coupling (direct interaction between the transmitter and receiver), which corresponds to the first ≈40 m when we use the 12.5m mode to study the fog layers.

The multi-wavelength microwave radiometer (MWR) HATPRO (Rose et al., 2005) is a passive remote sensing instrument that measures the downwelling radiation at 14 different microwave wavelengths at the surface. These radiances are inverted using an artificial neural network algorithm to estimate the vertical profiles of temperature and humidity of the atmosphere in the range 0–10 km and the total amount of liquid water in the atmospheric column (liquid water path, LWP, g $m^{-2}$). As the profiles are based on passive measurements, the vertical resolution is limited; however, in the boundary layer the measurements at different elevation angles enhance the resolution of the temperature profile, giving 4–5 degrees of freedom for the full temperature profile. The humidity profile only has about 2 degrees of freedom (Löhnert et al., 2009). The integrated water vapour (IWV) is more reliable with an uncertainty of $\pm0.2$ kg $m^{-2}$, while the estimate of LWP in general has an uncertainty of $\pm20$ g $m^{-2}$, according to the manufacturer. However, for small LWP (<50 g $m^{-2}$), investigations by Marke et al. (2016) indicate that the absolute uncertainties are smaller, with a root-mean-square (RMS) error of 6.5 g $m^{-2}$. Moreover, much of the uncertainty in retrieving LWP is due to uncertainties in atmospheric conditions, such as cloud temperature and humidity profile (e.g. Gaussiat et al., 2007), which usually will not change dramatically during one fog event. In absence of higher liquid clouds, the detection limit of changes in fog LWP should therefore be smaller, probably in the order of 5 g $m^{-2}$ (Bernhard Pospichal, personal communication). To reduce the constant bias in MWR LWP, we subtract the mean LWP retrieved during the 1-hour period of clear sky that is nearest in time to the fog event of interest. For the three fog events in 2014 studied in this paper (see Sect. 2.5), the imposed correction is 1.1, 5.2 and 23.9 g $m^{-2}$, respectively. An improvement of the instrument algorithm provided by the manufacturer in 2015 reduced this clear-sky bias to less than 1 g $m^{-2}$ for the rest of



the fog events. An approximate evaluation of the LWP uncertainty using LW radiation measurements suggests an RMS error in LWP of about 5–10 g m$^{-2}$ during fogs with LWP < 40 g m$^{-2}$ (Appendix A).

## 2.3 Radiation code ARTDECO

The radiative transfer is calculated using ARTDECO (Atmospheric Radiative Transfer Database for Earth Climate Observation), a numerical tool developed at LOA (Lille University) which gathers several methods to solve the radiative transfer equation and datasets (atmospheric profiles, optical properties for clouds and aerosols, etc.) for the modelling of radiances and radiative fluxes in the Earth's atmosphere under the plane-parallel assumption. Data and user guide are available on the AERIS/ICARE Data and Services Center website at http://www.icare.univ-lille1.fr/projects/artdeco. In this paper, the radiative transfer equation is solved using discrete-ordinate-method DISORT (Stamnes et al., 1988) in the solar spectrum (0.25–4 μm) and the thermal spectrum (4–100 μm). The spectral resolution is 400 cm$^{-1}$ in 0.25–0.69 μm, 100 cm$^{-1}$ in 0.69–4 μm and 20 cm$^{-1}$ in 4–100 μm, which gives 303 wavelength bands in total. Gaseous absorption by $H_2O$, $CO_2$ and $O_3$ is taken into account and represented by the correlated k-distributions (Dubuisson et al., 2005; Kratz, 1995). In ARTDECO, the coefficients of the k-distribution are calculated using a line-by-line code (Dubuisson et al., 2006) from the HITRAN 2012 spectroscopic database (Rothman et al., 2013). The use of correlated k-distribution makes it possible to account with accuracy for interaction between gaseous absorption and multiple scattering with manageable computational time. In addition, the impact of the absorption continua is modelled using the MT_CKD model (Mlawer et al., 2012). Optical properties of water clouds are calculated for a given droplet size distribution (DSD) using Mie calculations. In this study, the DSD is parametrized using a modified gamma distribution, applying parameter values presented by Hess et al. (1998) for fog and continental stratus. The effective radius is 10.7 μm for fog and 7.3 μm for stratus, but we modify the effective radius in the fog according to the radar reflectivity (see Sect. 3.2). Ice clouds are represented by the Baum & Co ice cloud parametrization implemented in the ARTDECO code (Baum et al., 2014), using an ice crystal effective diameter of 40 μm.

Radiative fluxes are calculated on 66 vertical levels spanning 0–70 km, 28 of which are located in the lowest 500 m in order to resolve fog layers well. A Lambertian surface albedo in the SW is applied, with a spectral signature representative of vegetated surfaces. However, as we observed that this albedo parametrization generally overestimates the observed albedo by ≈25 %, we downscale the albedo at all wavelengths to better fit the median albedo of 0.221 of October 2014–March 2015 observed at SIRTA. In the LW, a constant emissivity of 0.97 is used.

## 2.4 Calculation of radiation-driven liquid water condensation and evaporation

The radiation-driven condensation (or evaporation) rate is calculated assuming the air remains at saturation while cooling or warming from SW or LW radiation only, neglecting all adiabatic motions or mixing, but taking into account the latent heat of condensation. The derivations below are based on the thermodynamics of a saturated air parcel, which is described by e.g. Wallace and Hobbs (2006).





For N model levels at height $h_j$ (j=1,..,N), ARTDECO calculates the radiative heating rate in each of the N-1 layers between these levels, $\left(\frac{dT}{dt}\right)_{rad,j}$ (j=1,...N-1). We assume that if the j$^{th}$ layer contains cloud, its water vapour content will always be at saturation with respect to liquid water. To satisfy this, the condensation rate $C_{rad}$ due to the radiation must be:

$$C_{rad,j} = -\frac{d\rho_s}{dT}\left(\frac{dT}{dt}\right)_j, \tag{1}$$

where $\rho_s$ is the saturation vapour concentration (g m$^{-3}$), and $\frac{d\rho_s}{dT}$ its change with temperature. $\left(\frac{dT}{dt}\right)_j$ is the total air temperature tendency, which under the above assumptions equals the radiative heating rate plus the latent heat of condensation:

$$\left(\frac{dT}{dt}\right)_j = \left(\frac{dT}{dt}\right)_{rad,j} + \frac{L_v}{\rho_a c_p}C_{rad,j}, \tag{2}$$

where $L_v$ is the specific latent heat of condensation, $\rho_a$ the air density and $c_p$ the specific heat capacity of air at constant pressure. We estimate $\frac{d\rho_s}{dT}$ by combining the ideal gas equation for water vapour ($e_s = \rho_s R_v T$) and Clausius–Clapeyron's equation ($\frac{de_s}{dT} = \frac{L_v e_s}{R_v T}$), which yields:

$$\frac{d\rho_s}{dT} = \frac{e_s}{R_v T^2}\left(\frac{L_v}{R_v T} - 1\right), \tag{3}$$

where $R_v$ is the specific gas constant of water vapour, and $e_s$ is the saturation vapour pressure, which we estimate from the formula presented by Bolton (1980):

$$e_s(T) = 611.2 \exp\left(\frac{17.67\,(T-273.15)}{T-29.65}\right), \tag{4}$$

with $T$ in K and $e_s$ is Pa. Combining Eqs. (1) and (2), we get an expression for the radiation-driven condensation rate:

$$C_{rad,j} = -\frac{\frac{d\rho_s}{dT}}{1+\frac{L_v}{\rho_a c_p}\frac{d\rho_s}{dT}}\left(\frac{dT}{dt}\right)_{rad,j}. \tag{5}$$

We calculate this condensation rate for all layers within the fog and finally integrate in the vertical to obtain the total condensation rate in the whole fog (in g m$^{-2}$ h$^{-1}$), thus getting C$_{LW}$ and -E$_{SW}$.

It is worth noting that the gradient $\frac{d\rho_s}{dT}$ increases strongly with temperature. This implies that a warmer fog condensates more liquid water than a cold fog given the same radiative cooling rate. In fact, the condensed water per radiative heat loss increases almost linearly from 0.55 to 0.90 g m$^{-2}$h$^{-1}$ per W m$^{-2}$ when the fog temperature increases from -2 to 15 ˚C (not shown).



## 2.5 Overview of the analysed fog cases

We calculate the radiation at 15min intervals in seven fog events that occurred at SIRTA during the winter seasons 2014–2015 and 2015–2016. An overview of the atmospheric conditions during each of these fogs is given in Table 2. The fog events were chosen to cover an important range of variability in atmospheric conditions such as 2m temperature and IWV, as
well as fog properties such as geometric thickness and LWP, and we have included one fog event where cloud layers above the fog were observed. Considering all fog events at SIRTA in the winter seasons 2012–2016 with reliable LWP measurements from the MWR (e.g. excluding cases with liquid clouds above), in total 53 events, the 10[th], 25[th], 50[th], 75[th] and 90[th] percentile of the LWP distribution is 6.6, 16.4, 40.2, 68.0 and 91.2 g m$^{-2}$, respectively (not shown). The chosen fog events thus cover the typical range of fog LWP. Fog types can be defined by the mechanism of formation (Tardif and
Rasmussen, 2007). At SIRTA, radiation fogs and stratus-lowering fogs occur with about the same frequency, while other fog types are less common (Haeffelin et al., 2010; Dupont et al., 2016).

Fog presence is defined by the 10min average visibility at 4 m being below 1 km (American Meteorological Society, 2017). For a 10min block to be part of a fog event, the visibility should be below 1 km at least 30 min of the surrounding 50min period, based on the method proposed by Tardif and Rasmussen (2007), thus defining the fog formation
and dissipation time of each event. From this definition, fog event number 3 and 6 should each be separated into two events; however, we have chosen to regard them as single events because the cloud base lifts only a few tens of meters for 2–3 hours before lowering again.

## 3 Retrieval of geophysical properties

This section describes how the measurements at SIRTA are used to prepare the input data to the radiative transfer code:
profiles of cloud properties, temperature and humidity. Before they are used, the data from all the instruments, except the temperature and humidity profiles from the radiosonde and MWR, are averaged in a 10-minute block around the time of interest.

### 3.1 Fog and cloud boundaries

The fog or low stratus is searched for in the lowest 500 m of the atmosphere. Its cloud-base height is found using a threshold
value in the attenuated backscatter from the ceilometer of $2 \cdot 10^{-4}$ m$^{-1}$ sr$^{-1}$, following Haeffelin et al. (2016), or a 1 km threshold in visibility if the cloud-base height is close to the surface. The cloud-top height is set to the altitude where the 12.5m resolution radar data no longer detect a signal above noise levels. In very thin fog situations where the visibility at 20 m is above 1 km, indicating that the fog top is below 20 m, the cloud-top height is set to 10 m.

The presence and vertical extent of higher cloud layers is determined from the radar. The clouds are assumed to
extend over the gates where a signal is detected above the background noise.



### 3.2 Fog microphysical properties

We assume that the fog contains only liquid droplets, and no ice, which is a reasonable assumption as the screen temperature during the fogs studied here is -1 °C at lowest (Table 2) and ice crystals in fog rarely occur at temperature above -10°C (Gultepe et al., 2007). The optical properties of the fog then depend only the liquid water content (LWC) and the DSD. Only the extinction coefficient at 550 nm is required as model input in addition to the DSD, since ARTDECO can determine the optical properties at all 303 wavelengths by Mie calculations from this information (Sect. 2.3). The extinction coefficient of cloud droplets at visible wavelengths (including 550 nm) is well approximated by:

$$\alpha_{ext,visible} = \frac{3\ LWC}{2\varrho_l\ r_{eff}},$$ (6)

with LWC in g m$^{-3}$, $r_{eff}$ is the effective radius in μm and $\varrho_l$ the density of liquid water in g cm$^{-3}$ (Hu and Stamnes, 1993). The optical depth at visible wavelengths (OD) is obtained by integrating $\alpha_{ext,visible}$ in the vertical.

The 12.5m resolution mode of the radar is used to estimate LWC and $r_{eff}$ at each level in the fog. For liquid droplets, the backscattered radar signal is proportional to the sixth moment of the DSD, a quantity known as radar reflectivity $Z$:

$$Z = \int_0^\infty D^6 n(D) dD,$$ (7)

where $D = 2r$ is the droplet diameter and $n(D)dD$ is the number concentration of droplets with diameter between $D$ and $D + dD$. $Z$ has units mm$^6$ m$^{-3}$, but is usually expressed in units of dBZ, defined by $dBZ = 10 \cdot \log_{10}(Z)$. We have chosen to apply the empirical relationships of Fox and Illingworth (1997) relating the radar reflectivity $Z$ (dBZ) to LWC (g m$^{-3}$) and $r_{eff}$ (μm):

$$LWC = 9.27 \cdot 10^{0.0641\ Z}$$ (8)

$$r_{eff} = 23.4 \cdot 10^{0.0177\ Z}$$ (9)

These relationships were derived from aircraft measurements of the droplet spectrum in stratocumulus clouds, covering the range -40 dBZ to -20 dBZ. The relationships are not valid in the presence of drizzle, which strongly increases Z as droplets are larger. Drizzle presence typically occurs when Z > -20 dBZ (e.g. Matrosov et al., 2004). We therefore use the value of LWC and $r_{eff}$ obtained at Z = -20 dBZ for any higher Z. The relationships are plotted in Fig. 2.

LWC and $r_{eff}$ are estimated in each radar gate from cloud base to cloud top using these relationships, assuming no attenuation. For the lowest altitudes, where the radar data cannot be used, we apply the reflectivity of the lowest usable gate (usually at ≈50 m). The LWP of the MWR is then applied as a scaling factor to improve the estimate of LWC. This scaling is not performed if the MWR LWP is less than 10 g m$^{-2}$. If a higher cloud that may contain liquid is detected, the LWP should be partitioned between the fog and this cloud (see Sect. 4.3). Having obtained LWC and $r_{eff}$, the profile of $\alpha_{ext,visible}$ can thus be determined using Eq. (6). Below 30 m, we instead use the visibility measurements, which relate to visible extinction through Koschmieder's formula (e.g. Hautiére et al., 2006):



$$\alpha_{ext,visible} = -\frac{\ln 0.05}{\mathrm{Vis}} \approx \frac{3.0}{\mathrm{Vis}} \qquad (10)$$

Examples of the profiles of Z, LWC, $r_{eff}$ and $\alpha_{ext,visible}$ are shown in Appendix B. Uncertainties in the retrievals of microphysical properties are also discussed in Appendix B. To reduce the computational cost, only four different DSDs are given to the radiative transfer code, with effective radii of 4.0, 5.5, 8.0 and 10.7 μm, respectively. In one model run, the same

DSD is used at all altitudes, and it is selected by applying Eq. (9) on the vertical median of Z.

### 3.3 Profiles of temperature and gases

The radiation code requires the vertical profiles of temperature and the concentrations of the gaseous species ($H_2O$, $CO_2$, $O_3$) as input. For $CO_2$, a vertically uniform mixing ratio of 400 ppmv is used, while for $O_3$ we use the AFGL mid-latitude winter standard atmospheric profile (Anderson et al., 1986) which is provided in ARTDECO. This standard atmosphere is also used

for temperature and humidity (i.e. $H_2O$) above 20 km. Below 10 km, the temperature and humidity from the MWR is applied, while the previous radiosonde at Trappes is used in 10–20 km. The measured surface skin temperature is used for surface emission temperature, while the in situ measured air temperature is used in the 0–30 m layer. When there is no cloud base below 50 m, the MWR temperature profile is modified in the lowest 200 m of the atmosphere to gradually approach the temperature measured at 30 m.

Due to fog top radiative cooling and subsequent vertical mixing, the temperature profile is often characterized by a saturated adiabatic lapse rate inside the fog, capped by a strong inversion above the fog top (Nakanishi, 2000; Price et al., 2015). This vertical structure was also observed by the majority of the 12 radiosondes launched during four fog events in the ParisFog field campaign of 2006–07 (not shown). If a cloud base is present below 50 m, we therefore let the temperature decrease adiabatically with height from the measured value at the top of the mast, and then impose an inversion of 5 K per

100 m from the fog top until the temperature profile of the MWR is encountered. This inversion strength corresponds to what was typically observed by the aforementioned radiosondes. When a cloud base is present below 50 m, we also increase the humidity within the whole fog layer to saturation and decrease the humidity in the atmosphere above with the same integrated amount, thus improving the estimate of the humidity column above the fog top.

### 4 Results

We will now present the results obtained by applying the methodology described above to the seven fog events in Table 2. We first describe in some detail two contrasting fog events (Sect. 4.1), then we study the statistics of the radiative properties in all six fogs without clouds above (Sect. 4.2), and finally we study the impacts of the clouds appearing above the last fog event (Sect. 4.3).




## 4.1 Quantitative analysis of two contrasting fog events

Figure 3 shows the time series of several observed and calculated quantities during the fog event on 27 Oct 2014. The visibility and LWP time series (Fig. 3a) reveal that this fog has two distinct stages. From 02 UTC to 06 UTC, intermittent patches of very thin fog exist, seen from the fluctuating 4m visibility and the 20m visibility remaining well above the fog threshold. After 06 UTC, the fog develops in the vertical, causing the visibility at 20 m to drop. The fog grows to a thickness of about 100 m, as can be seen by the radar (Fig. 3b), reaching a maximum LWP of about 20 g m$^{-2}$ just after sunrise, at 07 UTC. A minimum visibility at 4 m (155 m) and at 20 m (87 m) is also reached at 07 UTC. After sunrise, the visibility steadily improves, fog dissipating at the surface at 08:50 UTC and nearly one hour later at 20 m.

Figure 3c–d shows the time series of temperature, wind speed and the net SW and LW downward radiation observed at 10 m. Before fog formation, the ground undergoes radiative cooling of $\approx$60 W m$^{-2}$, which gives rise to the observed strong temperature inversion in the first 20 m of the atmosphere. The surface radiation budget stays unchanged during the period of intermittent fog, indicating that the fog is restricted to below the 10m level where the flux is measured. Once the fog starts developing in the vertical, however, the 10m net LW radiation increases and becomes close to zero at the fog peak time at 07 UTC, indicating that the fog is nearly opaque to LW radiation at this time. In the same period, from 06 to 07 UTC, the stable temperature profile evolves into a nearly isotherm layer. After sunrise, strong SW absorption at the surface (reaching >100 W m$^{-2}$) is associated with a sharp rise in temperature, which likely explains the dissipation of the fog.

Figure 3e–h shows quantities that are calculated using our methodology. Until 06 UTC, the fog OD is based on the observed 4m extinction and an assumed thickness of 10 m, resulting in a very low fog OD. The estimated fog OD increases strongly from 06 to 07 UTC, reaching 4 at 07 UTC. This is associated with a distinct increase in downwelling LW at 10 m, qualitatively consistent with the observations (Fig. 3g). As the LW emissivity of the fog increases, the radiative cooling is transferred from the surface to the fog, causing an increase in the calculated $C_{LW}$, which reaches a maximum of 50 g m$^{-2}$ h$^{-1}$ (Fig. 3h). The magnitude of this parameter indicates that the radiative cooling process can produce the observed maximum in fog LWP is less than one hour, which is consistent with the observed increase in LWP. The underestimation of the downwelling LW at 10 m after 06 UTC can indicate that the calculated LW emissivity of the fog is slightly underestimated, and thus also $C_{LW}$. The calculation also underestimates the LW flux by about 15 W m$^{-2}$ before 06 UTC, which is probably due to uncertainties in the vertical profile of temperature and humidity (see Sect. 5.3). $E_{SW}$ is small, only $\approx$2 g m$^{-2}$ h$^{-1}$ (Fig. 3h). The heating of the fog via surface absorption is probably much more important for evaporating the fog.

Figure 4 shows the same quantities as Fig. 3, but for the fog event on 13 Dec 2015. In contrast to the fog on 27 Oct 2014, this fog forms from the gradual lowering of the cloud-base of a pre-existing low stratus, which is already much thicker than the fog on 27 Oct 2014. During the whole day, this fog has an LWP of 50–100 g m$^{-2}$ and a thickness of 250–300 m and thus remains optically thick. A transition from fog to low stratus occurs at 12:20 UTC, but the cloud base rises only to $\approx$20 m before descending again to form fog at 15 UTC (not shown). As the fog is opaque to LW, the good agreement between the modelled and observed downwelling LW at 10 m (Fig. 4g) reflects only the temperature of the fog. More interesting is the



good agreement between the modelled and observed downwelling SW radiation at 10 m (Fig. 4f), which indicates that the estimated fog OD is rather precise. $C_{LW}$ is around 50 g m$^{-2}$ h$^{-1}$ with little variability, indicating that this process can renew the fog LWP in 1–2 hours. $E_{SW}$ reaches 9 g m$^{-2}$ h$^{-1}$ around midday and is thus of less importance. This thicker fog also reflects more SW radiation than the fog 27 Oct 2014 so that less SW reaches the surface (Fig. 4f), which probably helps the fog to

persist, although the LWP decreases during the day.

## 4.2 Radiation-driven condensation and evaporation in six fogs without clouds above

Figure 5 shows the values of our three radiation parameters calculated every 15 minute during the six fog cases without higher clouds (Table 2). $C_{LW}$ varies significantly, from 0 to 70 g m$^{-2}$ h$^{-1}$ (Fig. 5a). Firstly, when the fog is not opaque to LW radiation, $C_{LW}$ is smaller, because the fog emits less than a blackbody. The optical depth of a cloud in the LW is principally

determined by its LWP (Platt, 1976). We therefore plot $C_{LW}$ against the MWR LWP in Fig. 5a, which shows that $C_{LW}$ increases strongly with LWP when LWP is smaller than 20–30 g m$^{-2}$. Remember, though, that the MWR LWP is not used in the input data to the radiation code when it is less than 10 g m$^{-2}$ (Sect. 3.2). When the fog is opaque (LWP $> \approx$ 30 g m$^{-2}$), the radiative cooling is restricted to the uppermost 50–100 m of the fog (Appendix B), in agreement with previous studies (Nakanishi, 2000; Cuxart and Jiménez, 2012). $C_{LW}$ then ranges 40–70 g m$^{-2}$ h$^{-1}$, varying significantly between fog events and

to a lesser degree ($\approx$5–15 g m$^{-2}$ h$^{-1}$) within the same event (Fig. 5a). Considering that the opaque fogs typically have an LWP of 30–100 g m$^{-2}$, the cooling by LW radiation constitutes a source of liquid water capable of renewing the entire fog in 1–2 hours, which is also the typical time scale for observed major changes in the fog LWP (not shown). The magnitude of $C_{LW}$ can be compared to the results of Nakanishi (2000), who studied the liquid water budget of fog in a large-eddy simulation. His Fig. 14a shows the domain-averaged profile of condensation rate in a 100m thick fog with LWP of about 15 g m$^{-2}$ (seen

from his Fig. 5b) in the morning. Condensation occurs in the upper 50 m of the fog, and the integral over these 50 m gives roughly 30–40 g m$^{-2}$ h$^{-1}$, which is similar to our results (Fig. 5a).

To investigate possible causes for the observed variability of $C_{LW}$ in opaque fogs, three situations with opaque fog (OD > 10) are compared in Fig. 6. $C_{LW}$ is 63.4, 47.7 and 61.6 g m$^{-2}$ h$^{-1}$, respectively (Fig. 6a). Since the fogs are opaque, the budget of LW radiation at the fog top is the main determining factor for the radiative cooling. Figure 6b shows the LW

fluxes at fog top in the three situations; the length of the vertical line indicates the net negative LW budget. The net LW budget is -73 W m$^{-2}$ both on 2 Nov 2015 and 8 Nov 2015, but the condensation rate is still 14 g m$^{-2}$ h$^{-1}$ higher on 8 Nov 2015. This is explained by the higher temperature of the fog top on the latter date (Fig. 6c), causing a higher condensation rate with the same cooling (see Sect. 2.4). The fogs on 28 Oct 2014 and 2 Nov 2015 differ in condensation rate by 16 g m$^{-2}$ h$^{-1}$. These two fogs have a very similar temperature, so the difference is explained by the LW radiative budget at fog top, which is -100 W m$^{-2}$ on 28 Oct 2014, i.e. 27 W m$^{-2}$ more negative than on 2 Nov 2015. This higher LW deficit can be

explained by the lower humidity above the fog (Fig. 6d) and possibly also the lower temperature in the first 1 km above the fog (Fig. 6c). Thus, $C_{LW}$ in the fogs without a cloud above varies significantly both from differences in fog OD, the fog temperature and the LW emission from the atmosphere above.



Figure 5b shows $E_{SW}$, which varies in 0–15 g m$^{-2}$ h$^{-1}$. $E_{SW}$ obviously depends on the amount of incoming SW radiation, so we plot it against the solar zenith angle. At one given angle, there is a variability of a factor of 4 between the fog cases. This variability is explained by the fog OD. Thinner fogs, such as on 27 Oct 2014 and 14 Dec 2014, will interact less with the SW radiation and therefore absorb less than the thicker fogs, such as on 28 Oct 2014 and 2 Nov 2015. $E_{SW}$ will

also depend on fog temperature through $\frac{d\rho_s}{dT}$, just like $C_{LW}$. All in all, $E_{SW}$ is generally much smaller than $C_{LW}$, even for thick fogs near (winter) midday, but it still represents a significant reduction in the net radiation-driven condensation rate in the fog in daytime relative to nighttime.

$R_{net,s}$ varies from 0 to 140 W m$^{-2}$ during daytime in the six fog cases (Fig. 5c). Absorption of SW is the dominant term, and therefore we highlight the dependency on solar zenith angle. However, net LW emission significantly reduces $R_{net}$

below the non-opaque fogs (27 Oct 2014 and 14 Dec 2014) with up to -60 W m$^{-2}$, and also frequently reaches -10 W m$^{-2}$ in the opaque fog situations because the ground is warmer than the fog (not shown). Since thicker fogs reflect more SW radiation, the absorbed SW is smaller below thick fogs than thin fogs at a given solar zenith angle, and this gives rise to the case-to-case variability in $R_{net,s}$ of a factor of 3 seen in Fig. 5c, e.g. from 40 W m$^{-2}$ to 120 W m$^{-2}$ at a solar zenith angle of 70˚. To study to what extent this absorbed heat is transferred to the fog, we compare the measurements of $R_{net,s}$ (at 10 m)

with the sensible heat flux measurements at 2 m in the fogs during daytime (Fig. 5d). The two parameters are clearly correlated. The fraction of sensible heat flux to $R_{net,s}$ in these data is found to have a 25 and 75 percentile of 0.20 and 0.40, respectively. Since 1 W m$^{-2}$ heating of the fog corresponds to an evaporation rate of about 0.7 g m$^{-2}$ h$^{-1}$ (Sect. 2.4), the sensible heat flux will cause an evaporation rate of roughly 0.15–0.30 g m$^{-2}$ h$^{-1}$ per W m$^{-2}$ of radiation absorbed at the surface. With a surface absorption of 100 W m$^{-2}$ at midday below thin fogs, this correspond to 15–30 g m$^{-2}$ h$^{-1}$ of

evaporation, which is almost as large as $C_{LW}$.

### 4.3 Radiation-driven condensation and evaporation in a fog with clouds above

Figure 7 presents the fog event occurring 1 January 2016, during which the BASTA cloud radar detects cloud layers appearing above the fog: traces of a stratus at ≈1.6 km from 07 to 08:30 UTC, and a higher and thicker stratus after 11 UTC. During the presence of the second cloud, the fog evaporates rapidly around 12–13 UTC, leaving only traces of a cloud at

≈150 m (Fig. 7b).

The radar mode at 200m resolution is just sensitive enough to detect the cloud at ≈1.6 km, so its geometrical thickness is uncertain. However, peaks in the LWP (Fig. 7a) appear at corresponding times when the cloud is observed by the radar. We therefore model the cloud as a liquid stratus and partition the LWP between the fog and overlying stratus cloud in the following way: In the period 06:45 to 07:30 (07:30 to 08:45) UTC, the first 30 (20) g m$^{-2}$ is attributed to the fog layer,

and the rest to the stratus. This results in an OD of the stratus of ≈10 when it is present (Fig. 7e). The stratus has a strong impact on $C_{LW}$ (Fig. 7h), reducing it by 90–100 %, because it increases the downwelling LW radiation at the fog top (not



shown). The presence of the stratus may therefore explain why the fog does not develop vertically, but instead decreases its geometric thickness and LWP while the stratus is present (Fig. 7a–b).

A second higher cloud appears at 11 UTC between 4 and 6 km. The cloud persists and deepens while the fog dissipates. From the radiosounding at ≈12 UTC, we know that the temperature in the 4–6 km layer is -25 to -13 ˚C. Since the LWP drops to zero after the fog cloud disappears, we choose to model the overlying cloud as a pure ice cloud, even though it is possible that it also contains liquid water while overlying the fog, which could explain the peaks in LWP around 12 UTC (Fig. 7a). To get a rough estimate of the OD of this cloud, we use an ice water content of 0.05 g m$^{-3}$, which corresponds to the average ice water content found by Korolev et al. (2003) for glaciated frontal clouds at temperatures of around -20 ˚C. This results in an OD of ≈5 in the beginning, growing with the observed thickness of the cloud (Fig. 7e). This cloud reduces $C_{LW}$ by ≈70 % (Fig. 7h), which is less than the effect of the first stratus. This is because the cloud is higher and colder, thus emitting less LW than the first cloud (Stephan Boltzmann's law). However, its effect is still more important than the variability in $C_{LW}$ found between cases without a higher cloud (Sect. 4.2). The cloud at 4 km also reduces $E_{SW}$ by 50–80 % and the SW reaching the surface by 15–30 %, due to reflection and absorption in the cloud, the effects increasing with time as the cloud thickens. Thus, in the SW the cloud has the opposite effect on the fog LWP as in the LW. However, the LW effect is more important than the SW effect for the fog LWP budget in this case: $C_{LW}$ decreases by ≈35 g m$^{-2}$ h$^{-1}$ due to the cloud presence, which is much more than the decrease in $E_{SW}$ of ≈4 g m$^{-2}$ h$^{-1}$ or the ≈10 W m$^{-2}$ reduction in the SW absorbed at the surface (not shown) which should correspond to less than 5 g m$^{-2}$ h$^{-1}$ decrease in evaporation by sensible heat flux (see Sect. 4.2).

The modelled and observed downwelling SW at 10 m are compared in Fig. 7f. They agree well both when there is only the fog (e.g. at 10 UTC), when both the fog and the cloud at 4 km are present (e.g. at 12 UTC) and when only the cloud is present (e.g. at 14 UTC), which provides a validation of the estimated OD of the fog and the cloud.

## 5 Discussion

We link the variability in the radiative parameters found in Sect. 4 to various properties of the atmospheric conditions, such as fog LWP and the presence or not of clouds above the fog. In order to understand better how each factor impacts the radiation-driven condensation and evaporation, theoretical sensitivity studies, where each input parameter is varied separately, are performed. Sensitivity to fog microphysical properties, temperature and humidity is analysed in Sect. 5.1, while impacts of higher clouds are explored in Sect. 5.2. Finally, a discussion of uncertainties is presented in Sect. 5.3.

### 5.1 Sensitivity of radiation-driven condensation and evaporation to fog properties, temperature and humidity

Figure 8 explores the sensitivity of our radiation parameters to the LWP and droplet sizes of the fog, which together determine its optical properties (see Sect. 3.2). The model runs use the input of the semi-transparent fog situation on 27 Oct 2014 at 08:30 UTC (Fig. 3), modifying only the fog LWP and/or the droplet effective radius.





Figure 8a shows that $C_{LW}$ increases fast with fog LWP when LWP is less than $\approx 30$ g m$^{-2}$. For higher LWP, the increase is much weaker, and beyond 50 g m$^{-2}$ it approaches a constant value as the emissivity of the fog approaches 1. The dependency on $r_{eff}$ for a given LWP is weak, which is due to a near cancellation between decreasing surface area and increasing absorption efficiency with $r_{eff}$, so that the LW optical depth of liquid clouds are almost entirely determined by LWP (Platt, 1976). The LW cooling process is thus sensitive to the fog LWP only if LWP $<\approx 40$ g m$^{-2}$, and it is not sensitive to droplet sizes within the range of effective radii studied here. Figure 8d shows that the downwelling LW flux at the surface increases with LWP in a very similar way as $C_{LW}$, which we use to evaluate the uncertainty in $C_{LW}$ due LWP uncertainty (Appendix A).

Figure 8b shows that $E_{SW}$ also increases with LWP. Compared to $C_{LW}$, $E_{SW}$ depends less strongly on LWP for thin fogs, but it keeps increasing with LWP also for opaque fogs with LWP well above 50 g m$^{-2}$. This is due to the SW radiation being largely diffused in the forward direction, rather than absorbed, so that there still remains much SW to be absorbed even far down inside an optically thick cloud. Note also that some absorption occurs even in when LWP=0, because of absorption by water vapour inside the cloud (Davies et al., 1984). $E_{SW}$ is also sensitive to the sizes of the droplets: for a given LWP, the largest effective radius (10.7 µm) gives a $\approx 50$ % larger evaporation rate than the smallest effective radius (4 µm), which can appear counter-intuitive since the total surface area of the DSD decreases with $r_{eff}$. This occurs due to an increase in absorptivity in the near-infrared with droplet size (Ackerman and Stephens, 1987).

The dependency of $R_{net,s}$ on fog properties (Fig. 8c) is the sum of LW and SW cloud effects. The fog reduces the SW reaching the surface by reflecting SW radiation, and this effect increases with LWP and decreases with $r_{eff}$ (Twomey, 1977). In the LW, radiative cooling of the surface is reduced as LWP increases, thus increasing $R_{net,s}$ with LWP, because the cooling is transferred to the fog top. Beyond LWP $\approx 40$ g m$^{-2}$, the sensitivity of $R_{net,s}$ to LWP is only due to SW. $R_{net,s}$ is about half as large when LWP is 100 g m$^{-2}$ than for LWP of 20 g m$^{-2}$. For thick fogs, the smallest droplets only let through half as much SW as the biggest droplets, while the dependency on droplet size is less pronounced for thin fogs.

In Fig. 9, we explore the sensitivity of $C_{LW}$ to the vertical profiles of temperature and humidity. In these tests, we use the opaque fog on 13 Dec 2015 at 10 UTC. Figure 9a confirms that an increase in fog-top temperature leads to a higher $C_{LW}$, by about 3 g m$^{-2}$ h$^{-1}$ per °C, caused both by higher emission of LW radiation by the fog (Stephan–Boltzmanns law) and the increase with temperature of the condensation rate per W m$^{-2}$ (Sect. 2.4). A temperature change in the atmosphere above the fog has a weaker impact of about 1.4 g m$^{-2}$ h$^{-1}$ per °C (Fig. 9c). Figure 9b illustrates that the first 100 m above the fog is in fact responsible for half of this effect, which is because most of the downwelling LW radiation under a cloud-free sky comes from the first few tens of meters, as noted by Ohmura (2001). The sensitivity to temperature above the fog is thus mainly related to the strength of the inversion at the fog top. The sensitivity of $C_{LW}$ to increased water vapour above the fog is about 2 g m$^{-2}$ h$^{-1}$ per added kg m$^{-2}$ of IWV (Fig. 9d), which confirms the importance of the dryness of the atmosphere found in Sect. 4.2.




## 5.2 Impact of radiation-driven condensation and evaporation on fog dissipation

The evolution of a fog depends on the competition between processes that produce liquid water and processes that remove it. Radiative cooling from emission of LW was found to be capable of producing 40–70 g m$^{-2}$ of liquid water per hour in the absence of a higher cloud layer, which is a significant source for maintaining the fog LWP, capable of renewing it in 1–2 hours against processes that remove LWP (deposition, turbulence). We found that $C_{LW}$ increases with fog temperature and decreases with the humidity in the overlying atmosphere; thus, warm fogs with a dry overlying atmosphere will be more resilient to dissipation than colder fogs with a more humid overlying atmosphere. However, these factors cannot be expected to vary very fast, so they will probably not be an initiating factor for the dissipation of a fog layer. On the contrary, the appearance of a second cloud layer above the fog can occur very fast by advection and instantly reduce $C_{LW}$ by several tens of g m$^{-2}$ h$^{-1}$ (Sect. 4.3). This should be sufficient to shift the balance in LWP in the direction of a fast reduction, leading to the dissipation of the fog.

In Fig. 10, we explore how a higher cloud affects the radiation-driven condensation and evaporation in an opaque fog as function of the OD and base altitude of the cloud. The impact on $C_{LW}$ (Fig. 10a) increases with the cloud OD, but beyond an OD of 5 this dependency is no longer very strong. The effect of the cloud weakens with increasing altitude of the cloud base; an opaque cloud at 10 km reduces $C_{LW}$ by only ≈30 %, while a cloud at 2 km reduces it by ≈100 %. This altitude dependency is due to the decrease of the temperature of the cloud with altitude due to the atmospheric lapse rate. At a given cloud OD and altitude, the effects of ice and liquid clouds are very similar. $E_{SW}$ is also reduced by the presence of a higher cloud (Fig. 10b), since the cloud absorbs and reflects the SW radiation that would otherwise be absorbed in the fog. It also decreases with OD of the cloud, while the altitude matters little. The decrease with cloud OD continues even for opaque clouds. However, beyond an OD of 5 it has already been more than halved and it decreases less rapidly. Since the fog in this case is opaque to LW, the cloud affects $R_{net,s}$ (Fig. 10c) mainly through its reflection of SW radiation, and the change is not dramatic since the fog is already reflecting most of the SW radiation. However, for a thin fog, $R_{net,s}$ is more strongly affected by the cloud, increasing due to the LW emission by the cloud and decreasing due to the SW reflection, similarly as it is affected by fog LWP for thin fogs in Fig. 8c (not shown).

The following conceptual comparison of the fog case on 13 Dec 2015 (Fig. 4) and the fog case on 1 Jan 2016 (Fig. 7) illustrates the possible role of radiation in determining the different evolutions of these two fogs. Both occur near mid-winter at a temperature of about 5 ˚C, and both are optically thick with LWP≈100 g m$^{-2}$ around midday (a). While the fog cloud dissipates completely right after midday on 1 Jan 2016, the fog on 13 Dec 2015 only slightly reduces its LWP during the afternoon, from ≈70 to ≈50 g m$^{-2}$. Based on the radiative transfer calculations, on 13 Dec 2015 $C_{LW}$ is ≈50 g m$^{-2}$ h$^{-1}$ and varies little, while on 1 Jan 2016 $C_{LW}$ is reduced from 50 g m$^{-2}$ h$^{-1}$ to 15 g m$^{-2}$ h$^{-1}$ when the higher cloud appears (h). The production of liquid water by LW cooling is thus 35 g m$^{-2}$ h$^{-1}$ higher in the fog on 1 Jan 2016 than in the fog on 13 Dec 2015, and the sink processes for liquid water must be stronger to dissipate the former. Conversely, the cloud also reduces the SW heating of the fog; at midday, $E_{SW}$ is ≈5 g m$^{-2}$ h$^{-1}$ less on 1 Jan 2016 compared to 13 Dec 2015, and the SW reaching the





surface is ≈40 W m$^{-2}$ less (f) (which means that the evaporation rate from sensible heat is likely ≈10 g m$^{-2}$ h$^{-1}$ less, see Sect. 4.2); however, this is less important than the difference in $C_{LW}$. Differences in other processes probably also play a role in the very different developments of the two fogs. For instance, the higher wind speed on 1 Jan 2016 (≈3 m s$^{-1}$, against 1–1.5 m s$^{-1}$ on 13 Dec 2015) could indicate that loss of liquid water by turbulent processes are more important on 1 Jan 2016 and

also contributes to its dissipation.

### 5.3 Uncertainty analysis

Table 3 provides rough estimates of the relative impact of the uncertainties in different measured and retrieved input data to the calculated values of $C_{LW}$, $E_{SW}$ and $R_{net,s}$. We assume that the uncertainties in these input data are more important than uncertainties related to the physics of the radiation model itself. The quantitative estimates are based on the results found in

the sensitivity studies, and on some further investigations that will be explained below.

Firstly, uncertainty arises from the estimates of fog optical properties. The uncertainty in fog LWP is found to be in the order of 5–10 g m$^{-2}$ when LWP < 40 g m$^{-2}$ (Appendix A). This corresponds to an uncertainty in $C_{LW}$ of 10–15 g m$^{-2}$ h$^{-1}$ (or 50 %) when LWP < 20 g m$^{-2}$ and 3–5 g m$^{-2}$ h$^{-1}$ (or 10 %) when LWP in 20–40 g m$^{-2}$ (Fig. 8a). $E_{SW}$ is affected both by the fog LWP and $r_{eff}$ (Fig. 8b). The estimated uncertainty in $r_{eff}$ of 30 % (Appendix B) indicates an uncertainty of ≈20 % in $E_{SW}$,

while the LWP uncertainty of ≈5–10 g m$^{-2}$ causes a similar uncertainty for small LWP, but lower for higher LWP (Fig. 8b). These uncertainties in LWP and $r_{eff}$ will also cause uncertainties in the order of 20–30 % in $R_{net,s}$, based on Fig. 8c. The uncertainties in $R_{net,s}$ is also estimated using the observed and modelled downwelling fluxes at 10 m, finding an RMS error of 0.046 in the SW transmissivity (translating to 20 W m$^{-2}$ SW absorption at solar zenith angle of 70˚), and an RMS error in the LW absorption of 13.8 W m$^{-2}$ when LWP < 20 g m$^{-2}$ and 4.8 W m$^{-2}$ when LWP is in 20–40 g m$^{-2}$ (Appendix A). Finally, it

should be noted that in the presence of a higher cloud containing liquid, the partitioning of LWP between the fog and this cloud will increase the uncertainty in the fog LWP.

Neglecting aerosols in the calculations is another source of uncertainty. While the scattering by aerosols will be small compared to that of the fog, additional in-fog heating by aerosol absorption of solar radiation can significantly increase $E_{SW}$, since multiple scattering by droplets increases the probability of absorption (Jacobson, 2012), and since the fog droplets

themselves only weakly absorb in the near-infrared. Previous studies (Chýlek et al., 1996; Johnson et al., 2004) have found that this increase in absorption is limited to ≈15 % in stratocumulus clouds. However, this effect might be enhanced in fog, since the aerosol concentration can get higher because the boundary layer is shallow and the fog is in direct contact with the surface. We test the impact of aerosols on $E_{SW}$ by adding two standard aerosol populations described by Hess et al. (1998) to the fog layer on 13 Dec 2015, with relatively low (0.05) and relatively high (0.15) aerosol optical depth at 550 nm (AOD)

(Table 4). The main difference between the two populations is that the urban aerosols include more black carbon particles than the continental average aerosols. Black carbon is responsible for most of the absorption, while its contribution to AOD is only 20 % and 6 % in the two populations, respectively. The resulting increase in $E_{SW}$ ranges from ≈10 % for continental average aerosols of AOD 0.05 to more than 100 % for urban aerosols with AOD 0.15 (



Table 4). Retrievals of AOD at SIRTA from a sun photometer, which requires direct sunlight and therefore has sparse temporal coverage, indicate that AOD is closer to 0.05 than 0.15 most of the time in October–March. Considering this, and that some aerosols will be located above the fog, the runs where AOD is set to 0.05 are the most realistic and show that the increase in $E_{SW}$ due to aerosols is probably not higher than 10–30 %. However, if black carbon optical depth

increases due to a strong pollution event, $E_{SW}$ could be more strongly enhanced. To investigate the aerosol effect on $E_{SW}$ in more detail, measurements of the aerosol chemical composition should be used in addition to the AOD, since the important parameter to estimate is the fraction of AOD represented by absorbing aerosols. Due to swelling of non-absorbing water soluble aerosols, this fraction is also impacted by the relative humidity at which AOD is measured. The interaction of the aerosols with the fog (e.g. immersion, wet deposition) can also modify their optical properties (Chýlek et al., 1996).

$C_{LW}$ has uncertainty related to the temperature and humidity profiles. As the screen temperature is known, fog temperature is more uncertain in opaque fogs than in thin fogs through the temperature difference between screen level and fog top. Since there is observational evidence that fog temperature profile is near adiabatic (Sect. 3.3), we assume that the uncertainty of the fog top is less than 1 ˚C even for very thick fogs, which should impact $C_{LW}$ less than 10 % (Fig. 9a). The MWR temperature profile has an uncertainty of less than 1 ˚C in the lower atmosphere (Löhnert and Maier, 2012), and even

with significant uncertainty in the shape of the temperature inversion above the fog, the sensitivity studies indicate that the impact on $C_{LW}$ is well below 10 % (Fig. 9b–c). The IWV of the MWR has an uncertainty of 0.2 kg m$^{-2}$ (Sect. 2.2), which corresponds to a very small uncertainty in $C_{LW}$ (Fig. 9d). However, as the vertical distribution of humidity is roughly estimated with only two degrees of freedom (Löhnert et al., 2009), sharp decreases in humidity, e.g. at the top of the boundary layer, will not be correctly represented. By analysing a case study where the humidity profiles from the radiosonde

and the MWR disagree strongly due to such a sharp decrease, we find an induced bias in $C_{LW}$ of less than 10 % ($\approx$4 g m$^{-2}$ h$^{-1}$).

We finally turn to the uncertainties related to the properties of the higher clouds. Firstly, as shown in Sect. 4.3, higher clouds may be undetected by the radar due to their low reflectivity. This is confirmed from non-fog situations, when the ceilometer often detects low stratiform clouds which affect significantly the downwelling LW at 10 m but that are

invisible to the radar (not shown). For the method of this paper to be reliable in cases where such thin clouds may occur, a more sensitive radar is required. According to Stephens et al. (2002), low level liquid clouds frequently have reflectivity down to -40 dBZ. The radar should therefore preferably have a sensitivity of -40 dBZ for all altitudes where liquid clouds occur ($\approx$1–6 km), even though it is probably less critical for mid-level clouds, which often contain some ice, which enhances their reflectivity. At high altitudes, thin cirrus clouds may also have reflectivity down to -40 dBZ, but those with reflectivity

below -25 dBZ rarely has OD > 1 (Stephens et al., 2002). Since high-level clouds with OD < 1 do not impact our results dramatically (Fig. 10), a sensitivity of -25 dBZ at high altitudes is acceptable.

Given that the higher cloud is detected, its altitude and thus temperature is readily estimated, so the uncertainty in its radiative impact is mainly related to its emissivity, which based solely on radar observations probably cannot be less uncertain than a factor of 2. If we are confident that the cloud is opaque (OD >$\approx$ 5), the uncertainty in its impact on $C_{LW}$ is





only a few g m$^{-2}$ h$^{-1}$, while a less opaque cloud will cause uncertainty of several tens of g m$^{-2}$ h$^{-1}$ (Fig. 10a). The relative uncertainty in E$_{SW}$ and R$_{net,s}$ caused by higher clouds are smaller than for C$_{LW}$ when the cloud is semi-transparent, but on the other hand it is also important for thick clouds (Fig. 10b–c). Finally, it should be noted that cases of fractional cloud cover also will cause uncertainty, since the radar only sees what appears directly above, while clouds covering only parts of the sky

also affect the radiation, in particular if they block the direct sunlight.

To conclude, the uncertainty in C$_{LW}$ is small ($\approx$10 %) when the fog is opaque (LWP $> \approx$ 30 g m$^{-2}$) and there either is no higher cloud or the higher cloud is opaque and covers all the sky, while a non-opaque fog and/or higher cloud will introduce higher uncertainty. A similar conclusion can be drawn for E$_{SW}$, although the uncertainty in the case of opaque fog/cloud remains higher than for C$_{LW}$, since the SW radiation penetrates deeper into the clouds than the LW cooling.

**6 Conclusions**

In this study, the magnitude and variability of the radiation-driven condensation and evaporation rate in continental fogs during mid-latitude winter have been quantified. Based on the results of this study, Table 5 summarizes how different atmospheric conditions will impact the susceptibility of a fog to dissipation by affecting the radiative processes.

Firstly, the cooling of the fog by emission of LW radiation provides an important source of liquid water. In opaque
fogs (LWP $> \approx$ 30 g m$^{-2}$) without an overlying cloud layer, this cooling seen in isolation will cause 40–70 g m$^{-2}$ h$^{-1}$ of condensation, which means that the fog typically can renew its liquid water in 1–2 hours through this process. Its variability can mainly be explained by fog top temperature and the humidity above the fog, with warmer fogs below drier atmospheres producing more liquid water. In thin fogs, the condensation is weaker, and the estimate is more uncertain due to the uncertainty in LWP of the fog.

The solar radiation absorbed by fog droplets causes a radiative heating of the fog layer during daytime. This heating decreases with solar zenith angle and increases with droplet effective radius and fog LWP. At (winter) midday, the evaporation rate from this heating can reach 15 g m$^{-2}$ h$^{-1}$ in thick fogs, while it is weaker for thin fogs (0–5 g m$^{-2}$ h$^{-1}$), based on absorption by pure liquid droplets only. The role of absorbing aerosols in fog is not extensively studied in this paper, but our results indicate that it increases the absorption of solar radiation by 10–30 % in a typical airmass at SIRTA. This aerosol
absorption effect can be worth investigating in more detail using observations of aerosol chemical composition, as it could be stronger during pollution events. The important parameter is the optical depth of the absorbing aerosols, which might be only a small fraction of the total aerosol optical depth.

The radiative heating of the surface in daytime is more important for thin fogs than thick fogs, and it is found to vary from 40 to 140 W m$^{-2}$ at a solar zenith angle of 70˚ from the thickest to the thinnest fog studied here. In situ
observations indicate that on average 20–40 % of this energy is transferred to the fog as sensible heat. Since 1 W m$^{-2}$ heating of the fog corresponds to an evaporation rate of $\approx$0.7 g m$^{-2}$ h$^{-1}$, this process can cause an evaporation rate of up to 30 g m$^{-2}$ h$^{-1}$



when the sun is high and thus likely be very important for reducing the LWP of the fog. A more detailed investigation of the surface energy budget during fog could lead to a more precise quantification the evaporation of fog by sensible heat.

The appearance of a second cloud layer above the fog strongly reduces the LW cooling of the fog, especially a low cloud. The LW-induced condensation rate can be reduced by 100 % if the low cloud is optically thick, and even by more

than 50 % for a semi-transparent cloud of optical depth 1. The presence of an overlying cloud can therefore be a determining factor for fog dissipation as the fog will then have much of its production of liquid water cut off. In cases where no cloud appears above the fog it is unlikely that the LW cooling can change fast enough for it to be a determining factor for the dissipation. The detection of clouds above the fog with the cloud radar is therefore crucial for analysing the impact of radiative processes on fog dissipation. To detect all important clouds above the fog, the radar sensitivity must be sufficient to

capture thin water clouds, requiring a sensitivity of -40 dBZ in the lower troposphere, and optically important high clouds, requiring a sensitivity of -25 dBZ in the upper troposphere. Current generation BASTA radars, which have a sensitivity of -40 dBZ up to 4 km and -30 dBZ at 10 km, should be able to detect most of the important clouds.

The results of this paper have been obtained from the use of multiple instruments, in particular cloud radar, ceilometer and microwave radiometer. If these measurements can be rapidly transferred and processed, the methodology of

this paper could be applied to quantify the radiation-driven condensation and evaporation rates in the fog in real-time, to be used to support short-term fog forecast. In order to be less instrumentally demanding and thus more applicable to other sites, a simplified method using only the cloud radar and ceilometer could be envisaged, supplemented by screen temperature and visibility measurements and integrated water vapour from a GPS. Even though LWP will be less accurately estimated without the microwave radiometer, this method would still be able to capture the most important factors: higher cloud

presence, fog vertical extent, fog temperature and integrated water vapour.

**Data availability**

Radar, ceilometer and radiosonde data as well as the measurements of radiative fluxes at 10 m, surface meteorological parameters and visibility are available from the SIRTA public data repository, which is accessible online at http://www.sirta.fr. The data policy and a data download are available from the website. The data from the MWR and the

data used for calculating the sensible heat fluxes are available on request on the SIRTA web site: http://sirta.ipsl.fr/data_form.html. The data and code of ARTDECO are available on the AERIS/ICARE Data and Services Center website: http://www.icare.univ-lille1.fr/projects/artdeco.

**Appendix A: Validation of surface radiative fluxes and LWP using radiation measurements at 10 m**

Figure A1a evaluates the accuracy of the modelled downwelling SW fluxes at 10 m with the observed fluxes during the six

fogs without higher cloud (Table 2). To eliminate the dependency on solar zenith angle, the fluxes are normalized with the



incoming flux at the top of the atmosphere; we thus validate the atmospheric SW transmissivity. The disagreements between the observed and modelled transmissivity is mainly caused by uncertainty in the fog opacity. The RMS error was found to be 0.046, and the spread is similar for different values of transmissivity (Fig. A1a). This corresponds to an uncertainty in the downwelling SW at the surface of about 20 W m$^{-2}$ when the solar zenith angle is 70˚.

We validate the downwelling LW flux at the surface when modelled fog LWP < 20 g m$^{-2}$ and when it is 20–40 g m$^{-2}$, respectively (Fig. A1b). In this LWP range, the fog is not yet completely opaque to LW radiation, so that the downwelling LW at the surface increases with fog LWP, typically by several tens of W m$^{-2}$ in the range 0–40 g m$^{-2}$ in absence of higher clouds (Fig. 8d). Because the disagreement between modelled and observed surface clear-sky downwelling LW at the surface is typically no more than 5–15 W m$^{-2}$ (based on two days of clear sky, not shown), the disagreement between

modelled and observed downwelling LW flux below a non-opaque fog with no higher clouds will mainly be due to the error in fog LWP. Thus, the validation of the surface downwelling LW flux can be used to estimate the uncertainty in LWP. Since the LWP dependency of the downwelling LW flux at the surface is very similar to the LWP dependency of $C_{LW}$ (Fig. 8ad), we are also able to estimate the uncertainty in $C_{LW}$ related to fog LWP. Based on the six fog events without higher clouds (Table 2), we find an RMS of the difference between observed and modelled downwelling LW flux at the surface of 13.8 W

m$^{-2}$ when the (estimated) LWP < 20 g m$^{-2}$ and 4.8 W m$^{-2}$ when LWP is 20–40 g m$^{-2}$. This corresponds to about 5–10 g m$^{-2}$ of uncertainty in LWP in both cases, considering Fig. 8d, which would cause roughly 10–15 g m$^{-2}$ h$^{-1}$ uncertainty in $C_{LW}$ for LWP < 20 g m$^{-2}$ and 3–5 g m$^{-2}$ h$^{-1}$ for LWP 20–40 g m$^{-2}$ (Fig. 8a), which is a relative uncertainty of 50 % and 10 %, respectively.

**Appendix B: Estimation of vertical profiles of microphysical and radiative properties in fog**

The method used in this study for relating the radar reflectivity Z to microphysical properties (Sect. 3.2) is only one of many possible approaches. The relationships can be derived by assuming a theoretical shape of the DSD (e.g. Maier et al., 2012), from a purely empirical fit to measurements from field campaigns (Fox and Illingworth, 1997; Sauvageot and Omar, 1987) and by modelling of microphysical processes (Khain et al., 2008). However, accurate and general relationships cannot be found from Z alone, since Z is most sensitive to the largest droplets, which may only weakly impact LWC and r$_{eff}$. As the

shape of the DSD varies significantly during and between fog events (Boers et al., 2012; Gultepe et al., 2007; Price, 2011), retrievals of LWC and r$_{eff}$ using Z alone will only be rough estimates, even in the absence of drizzle. A synergy with the more reliable LWP from MWR is therefore used in several methods in the literature, with varying approaches for vertically distributing this liquid water inside the cloud. For example, the LWC can be assumed to increase linearly with height due to sub-adiabatic up- and downdrafts (e.g. Boers et al., 2000). More complex algorithms to retrieve LWC and r$_{eff}$ which also

utilize the ceilometer extinction (e.g. Martucci and O'Dowd, 2011) or the radar Doppler velocity (e.g. Kato et al., 2001) have also been developed.





Comparisons during 25 fog events observed at SIRTA reveal that the LWP estimated from Eq. (8) is often a factor 2–3 smaller than the MWR LWP (not shown). However, since we normalise the LWC with the MWR LWP, only the vertical distribution of LWC is impacted by the Z–LWC relationship, except when LWP $< 10$ g m$^{-2}$. This vertical distribution will not strongly impact our main results, since they are based on vertically integrating throughout the fog. On the other hand, the uncertainty in $r_{eff}$ remains and will impact the calculated optical properties of the fog. The results of Fox and Illingworth (1997) indicate that the estimate of $r_{eff}$ from Z comes with an RMS error of about 20 %. Using the optical particle counter LOAC (Renard et al., 2016) lifted by a tether balloon during a few hours of a fog event at SIRTA when Z varied from -40 to -20 dBZ, we found a Z–$r_{eff}$ relationship similar to Eq. (9), even though $r_{eff}$ was ≈25 % smaller (not shown). Although only based on one case, this still indicates that Eq. (9) is an acceptable estimate for $r_{eff}$ in fog, and that the uncertainty in $r_{eff}$ is roughly in the order of 30 %. Finally, a calibration uncertainty of the radar of $1-2$ dBZ also impacts the retrieval of LWC and $r_{eff}$, but it is apparent from Fig. 2 that the impact of this uncertainty is less important than the uncertainties in relating Z to LWC and $r_{eff}$.

Figure B1a–d shows some examples of the vertical profiles of microphysical properties in the fog calculated using the method of our study, for one thin and two thick fog situations. The observed profile of Z typically has a maximum somewhere in the middle of the fog and decreases towards the bottom and top, as seen in Fig. B1a. This therefore translates into profiles of LWC, $r_{eff}$ and visible extinction with a similar shape (Fig. B1b–d). The visibility meters indicate that the extinction decreases strongly on approaching the surface (Fig. B1d). This vertical gradient in extinction is probably related to evaporation and deposition of fog droplets near the surface, which means that the LWC is probably in reality also decreasing strongly on approaching the surface, in continuation of the decrease observed above the radar blind-zone in the thick fog situations (Fig. B1b). Compared to methods assuming a linear increase of LWC with height, our method usually produces a stronger vertical gradient in LWC in the lower fog and a lower LWC near the fog top, with the level of maximum LWC often significantly below the fog top.

The LW radiative cooling occurs predominantly in the first 50 m below fog top (Fig. B1e), as also found in modelling studies of fog (Nakanishi, 2000). The peak cooling rate is stronger and more vertically restricted in the thick fogs than in the thin fog, due to the extinction coefficient near the fog top being higher (Fig B1d). The extinction coefficient in the thin fog may be underestimated, though, since the MWR LWP is not used to scale the fog LWP in this case (as MWR LWP $< 10$ g m$^{-2}$). Near the surface, there is radiative heating when the surface is warmer than the fog. This occurs in all the shown cases. In absence of solar radiation on 28 Oct 2014 at 04:30 UTC, this warmer surface can be explained by the fog being cooled from above while the ground is sheltered by the fog. The SW heating rate (Fig. B1f) is also strongest near the fog top, but it penetrates further down into the fog than the LW cooling, which can be explained by the strong forward scattering by droplets and also agrees with the results of Nakanishi (2000). The SW heating rate is also significant above the fog, due to molecular absorption (dominantly by water vapour), which indicates that water vapour absorption inside the fog can also be important for heating the fog, as discussed by e.g. Davies et al. (1984). Finally, the calculated condensation rates (Fig. B1g–h) show the same patterns as the radiative heating rates with opposite sign, as expected. Condensation is occurring



mainly near the fog top due to LW cooling, while a weaker evaporation is induced in the lower parts of the fog from SW and LW heating.

## Author contribution

E. Wærsted carried out the radiation simulations, supervised by M. Haeffelin and J.-C. Dupont.

J.-C. Dupont calculated the surface sensible heat fluxes.

P. Dubuisson is a developer of the ARTDECO code. He helped E. Wærsted carry out the simulations.

J. Delanoë is a developer of the cloud radar BASTA. He deployed it at the SIRTA observatory and helped with the interpretation of the radar data.

E. Wærsted and M. Haeffelin prepared the manuscript, with contributions from all co-authors.

## 10   Competing interests

The authors declare that they have no conflict of interest.

## Acknowledgements

This research is supported financially by the French ministry of defence – Direction Gérérale de l'Armement, and by the company Meteomodem. The authors would like to acknowledge SIRTA for providing the observational data used in this
study. We thank the ICARE Data and Services Center for providing access to the ARTDECO model with associated datasets. ARTDECO has been developed with a financial support (TOSCA program) from the French space agency, "Centre National d'Etudes Spatiales" (CNES). We thank Jean-Baptiste Renard for his help with the instrument LOAC, enabling us to measure the droplet size distribution in fog.

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





**Table 1: Vertical and temporal resolution of the observations used in this study. All instruments are located at the SIRTA observatory main facility, apart from the radiosondes which are launched at Trappes (15 km west of the site) at approximately 11:15 and 23:15 UTC. The measurements by the cloud radar, ceilometer and microwave radiometer are obtained from remote sensing, while the other instruments measure in situ.**

| Instrument | Measured quantity | Vertical range and resolution | Temporal resolution |
|---|---|---|---|
| Cloud radar BASTA | reflectivity (dBZ) | RA 0–6 km, RE 12.5 m | 12 s |
| | | RA 0–12 km, RE 25 m | |
| | | RA 0–12 km, RE 100 m | |
| | | RA 0–12 km, RE 200 m | |
| Microwave radiometer | Liquid water path (g m$^{-2}$) | Integrated | 60 s |
| | Temperature profiles (K) | RA 0–10 km, 4–5 degrees of freedom | ≈5 min |
| | Humidity profile (g m$^{-3}$) | RA 0–10 km, 2 deg. of fr. | ≈5 min |
| Ceilometer CL31 | Attenuated backscatter | RA 0–7.6 km, RE 15 m | 30 s |
| Visibility meters | Visibility (m) | At 4 m, 20 m | 60 s |
| Thermometers on 30m mast | Air temperature (K) | At 1, 2, 5, 10, 20, 30 m | 60 s |
| Thermometer (unsheltered) | Surface skin temperature (K) | At ground level | 60 s |
| Cup anemometer | Wind speed (m s$^{-1}$) | At 10 m | 60 s |
| CSAT-3 sonic anemometer and LI-7500 infrared gas analyser | Sensible heat flux and latent heat flux (W m$^{-2}$) | At 2 m | 10 min |
| Radiosondes | Temperature (K) and humidity (g m$^{-3}$) profiles | RA 0–30 km, RE ≈ 5 m | 12 h |
| Pyranometers | Down- & upwelling irradiance in the solar spectrum (W m$^{-2}$) | At 10 m | 60 s |
| Pyrgeometers | Down- & upwelling irradiance in the terrestrial spectrum (W m$^{-2}$) | At 10 m | 60 s |



**Table 2: Main characteristics of each fog event studied in this paper. Dissipation time is relative to sunrise (–: before, +: after). The fogs are classified as radiation fog (RAD) or stratus-lowering fog (STL), as defined by Tardif and Rasmussen (2007). Pressure is measured at 2 m and is indicated for the time of formation, while the bracketed value indicates how much higher (+) or lower (-) the pressure is 24 h later.**

| No | Time of formation (UTC) | Duration (hh:mm) | Diss. time rel. to sunrise (h) | Fog type | Pressure (hPa) | Higher clouds (y/n) | Min. visi (m) at 4 m | Median (Max) LWP (g m$^{-2}$) | Max thickness (m) | 2m temp. range (°C) | IWV range (kg m$^{-2}$) |
|----|------------------------|------------------|-------------------------------|----------|----------------|---------------------|----------------------|-------------------------------|-------------------|---------------------|--------------------------|
| 1 | 27 Oct 2014 04:30 | 4:20 | +2.3 | RAD | 1006(-5) | n | 135 | 6 (22) | 110 | 7.2–9.4 | ≈9–13 |
| 2 | 28 Oct 2014 00:50 | 8:20 | +2.5 | RAD | 1001(-3) | n | 145 | 130 (209) | 450 | 7.0–9.8 | ≈7–9.5 |
| 3 | 14 Dec 2014 06:00 | 17:10* | –8.6 | RAD | 999(+0) | n | 103 | 18 (56) | 210 | (-1.1)–2.5 | ≈6–9 |
| 4 | 2 Nov 2015 05:00 | 9:20 | +7.6 | RAD | 1007(-8) | n | 74 | 62 (105) | 275 | 5.1–8.5 | ≈9–11 |
| 5 | 8 Nov 2015 05:50 | 4:00 | +2.9 | RAD | 1009(-1) | n | 128 | 40 (61) | 210 | 13.7–14.4 | ≈22–28 |
| 6 | 13 Dec 2015 06:20 | 29:20* | +3.9 | STL | 1003(-3) | n | 72 | 69 (135) | 360 | 2.8–5.7 | ≈10–14 |
| 7 | 1 Jan 2016 07:00 | 5:20 | +4.5 | RAD | 1006(-17) | y | 125 | 67 (154) | 410 | 4.6–5.9 | ≈12–15 |

5  *the cloud base lifted to a few tens of meter on 14 Dec 2014 during 13:40–15:10, and on 13 Dec 2015 during 12:20–15:00.

**Table 3: Rough estimates of the relative uncertainty (in % of the estimated value) of each radiation parameter (defined in Sect. 2.1), due to various sources of uncertainty, for thin (LWP ≪ 30 g m$^{-2}$) and thick (LWP > 30 g m$^{-2}$) fog situations. The (second-)last row is relevant when an opaque (semi-transparent) cloud overlies the fog. See text for details.**

| Uncertainty source | $C_{LW}$ | | $E_{SW}$ | | $R_{net,s}$ (day) | |
|--------------------|----------|-------|----------|-------|-------------------|-------|
| | Thin | Thick | Thin | Thick | Thin | Thick |
| Fog LWP | 10–50* | <10 | 20–40* | 10 | 10 | 10 |
| Droplet effective radius | <5 | <5 | 20 | 20 | 20 | 30 |
| Neglecting absorbing aerosols | – | – | 10–30[p] | 10–30[p] | <5 | <5 |
| Temperature profile | 5 | 5–10 | – | – | – | – |
| Humidity profile | 5–10 | 5–10 | – | – | – | – |
| OD of semi-transparent cloud above | 20–80** | 20–80** | 50–80 | 50–80 | 30 | 20 |
| OD of opaque cloud above | <10 | <10 | 50 | 50 | 30 | 20 |

10  [p] Uncertainty towards higher values only

*Uncertainty is highest for the thinnest fogs.

**Uncertainty is bigger for low clouds than high clouds.





**Table 4: Effect on $E_{SW}$ (defined in Sect. 2.1) by adding aerosols to the fog layer on 13 Dec 2015 at 12 UTC. Urban and continental average aerosols are defined as in Hess et al. (1998). The aerosol optical depth (AOD) is spread evenly across the 275m thick fog layer.**

| Type of aerosol | Aerosol single scattering albedo at 550 nm, at 80 % relative humidity | AOD at 550 nm, at 80 % relative humidity | $E_{SW}$ (g m$^{-2}$ h$^{-1}$) |
|---|---|---|---|
| No aerosols | – | 0 | 7.9 |
| Urban | 0.817 | 0.05 | 11.0 |
| | | 0.15 | 16.5 |
| Continental average | 0.925 | 0.05 | 8.8 |
| | | 0.15 | 11.5 |

**Table 5: Summary of how the susceptibility of fog to dissipation is affected by variability in atmospheric conditions through radiative processes. "Positive" ("negative") means that the fog is more (less) likely to dissipate due to lower (higher) net production of liquid water by the indicated radiative process (defined in section 2.1) due to the indicated atmospheric property. See text for details.**

| Atmospheric property | Less LW-driven condensation ($C_{LW}$) | More SW-driven evaporation ($E_{SW}$) | More surface heating ($R_{net,s}$) |
|---|---|---|---|
| Clouds above fog | strongly positive | negative | negative |
| Low fog LWP ($< 30$ g m$^{-2}$) | strongly positive | negative | positive |
| Absorbing aerosols in fog | – | positive | – |
| Higher fog temperature | negative | weakly positive | weakly positive |
| More humidity in atmosphere above fog | positive | – | – |
| Stronger temperature inversion above fog | weakly positive | – | – |





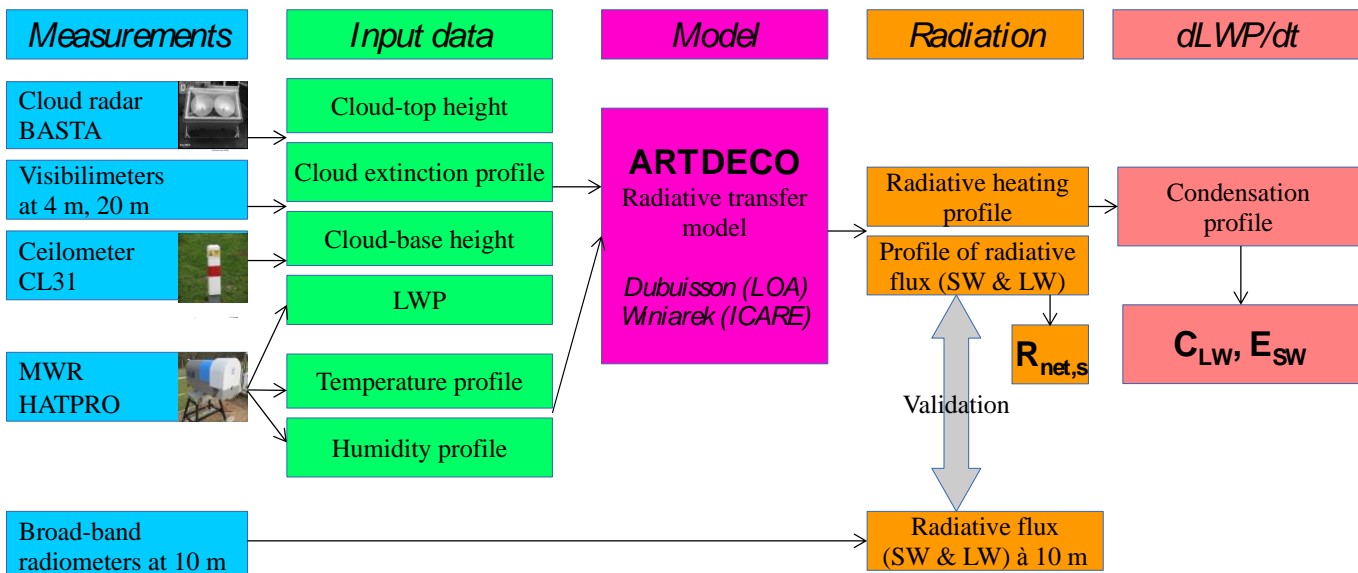

**Figure 1: Schematic overview of the methodology.**

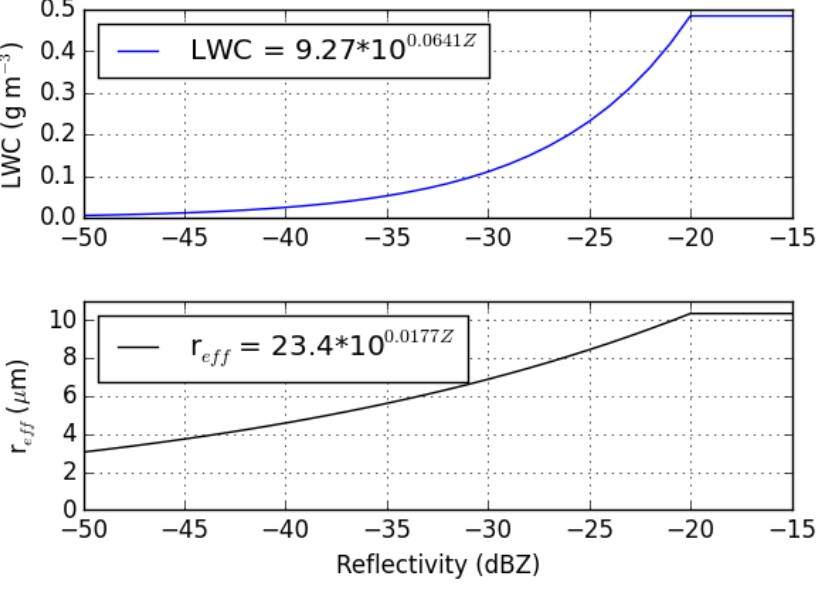

**Figure 2: Empirical relationships between radar reflectivity (Z) and LWC and effective radius used in this study, modified from Fox and Illingworth (1997).**



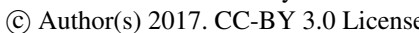


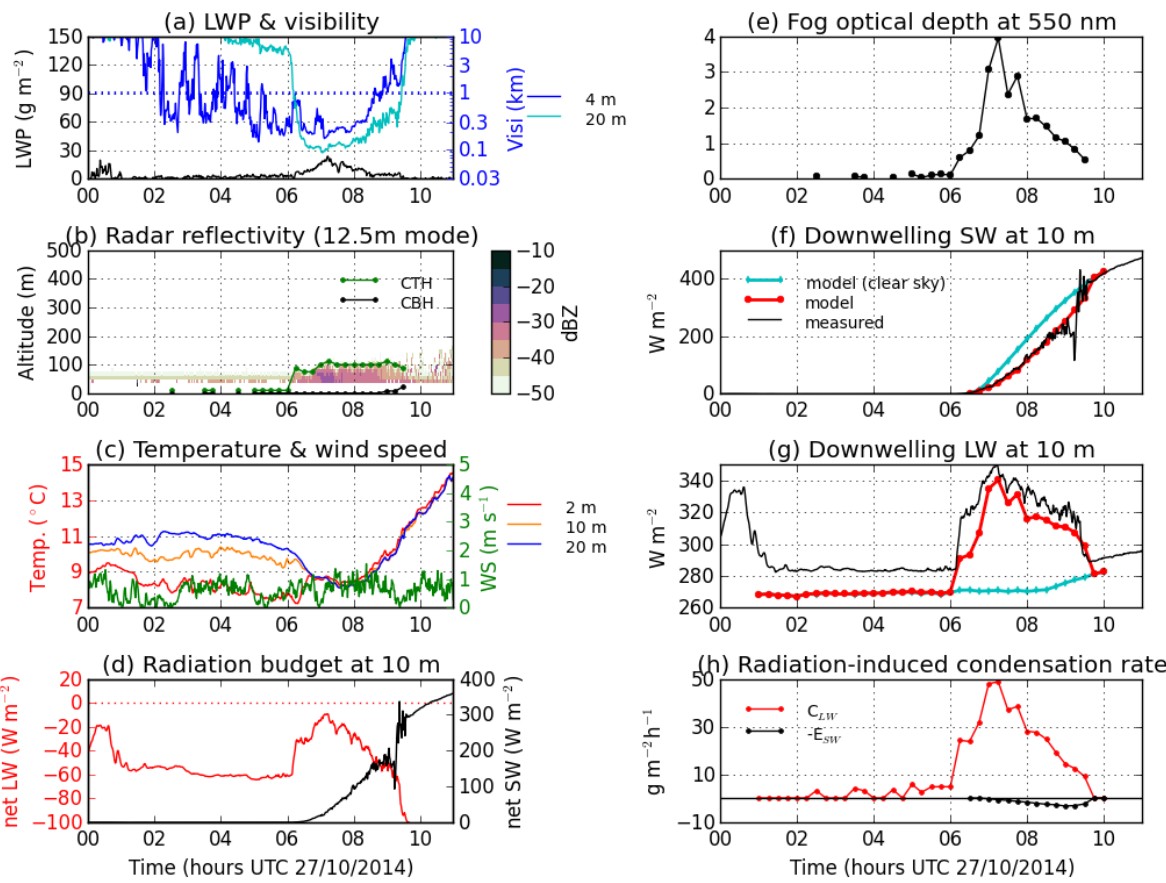

**Figure 3: The fog event on 27 Oct 2014. (a–d) Time series of observed variables: (a) LWP from MWR (g m$^{-2}$) and visibility (m) at 4 m and 20 m; (b) profile of radar reflectivity (dBZ), and estimated cloud-base height (CBH) and cloud-top height (CTH); (c) temperature (°C) at 2 m, 10 m and 20 m, and wind speed (m s$^{-1}$) at 10 m; (d) net downwelling SW and LW radiative flux (W m$^{-2}$) at 10 m. (e–h) Time series of calculated variables: (e) fog optical depth at 550 nm; (f) downwelling SW flux (W m$^{-2}$) at 10 m, comparing model runs including the fog, model runs not including the fog (clear sky) and the measurement; (g) as f, but for the downwelling LW flux; (h) the vertically integrated condensation rates (g m$^{-2}$ h$^{-1}$) due to LW and SW radiation ($C_{LW}$ and $E_{SW}$, defined in Sect. 2.1).**

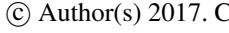

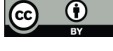

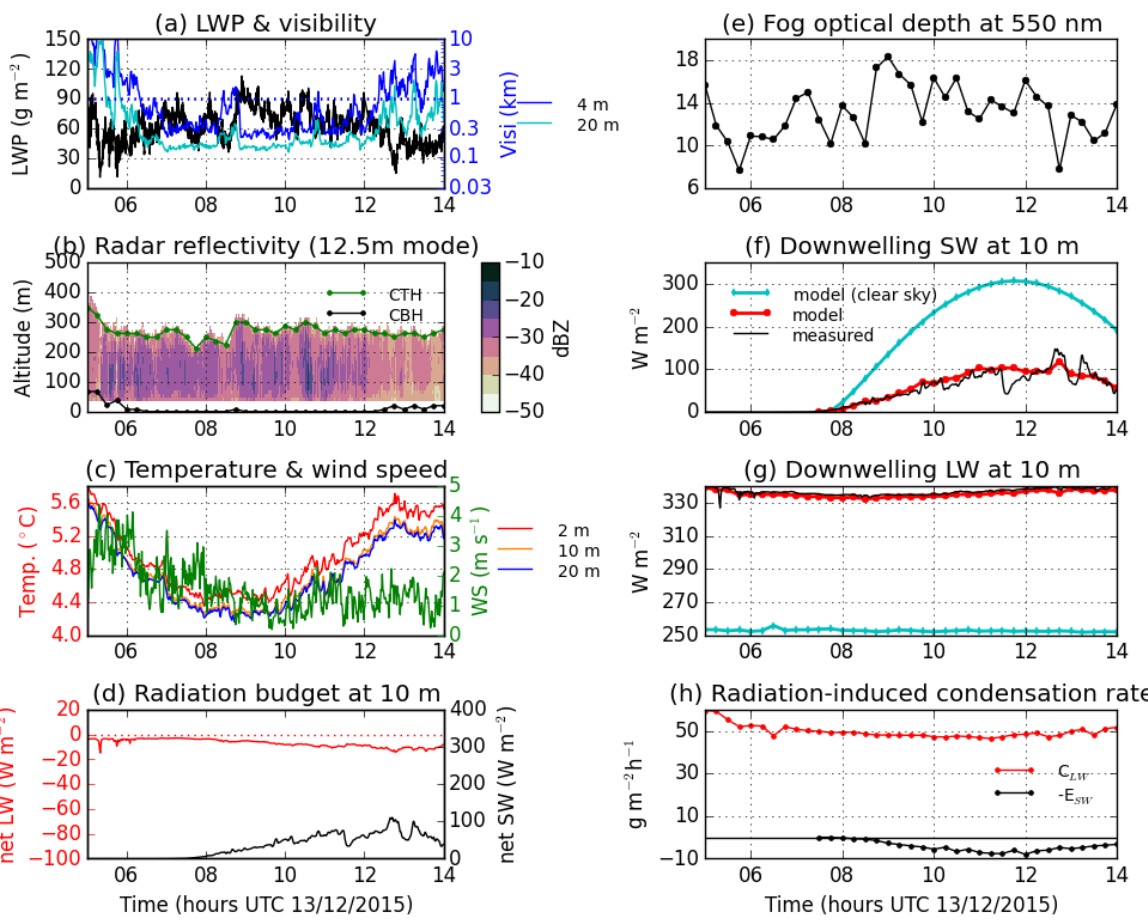

**Figure 4: Same as Fig. 3, but for the fog event on 13 Dec 2015.**



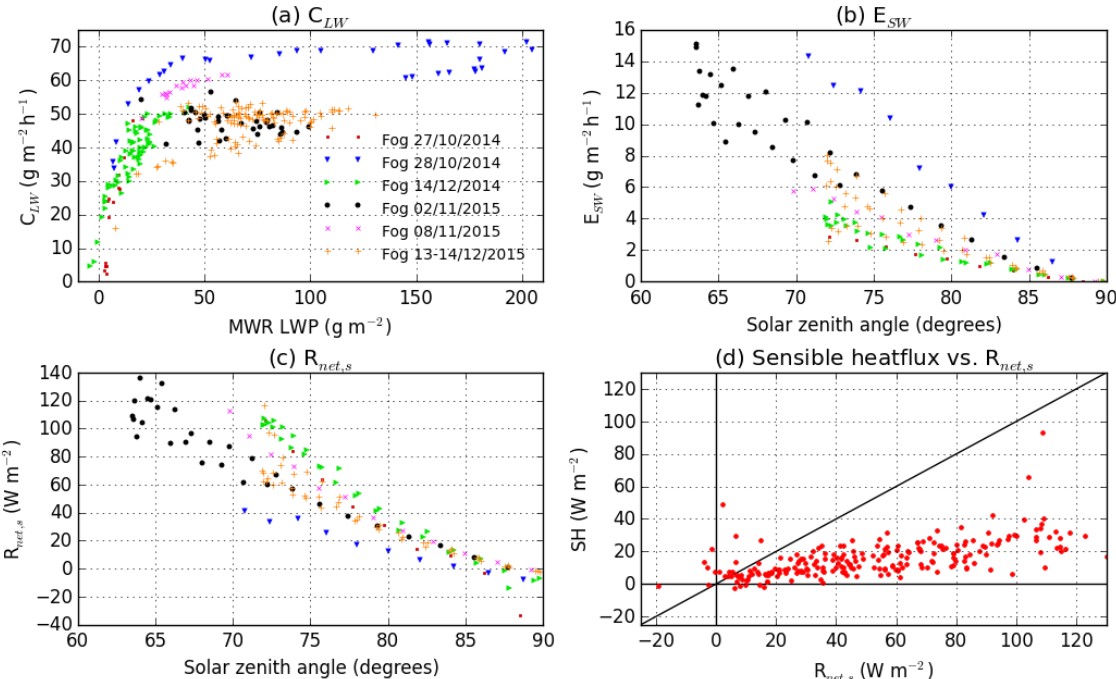

**Figure 5:** $C_{LW}$ (a), $E_{SW}$ (b) and $R_{net,s}$ (c) (defined in Sect. 2.1), calculated every 15 minute from formation time to dissipation time for the six fogs without clouds above in Table 2. (d) Measured 10min average sensible heat flux at 2 m vs. measured 10min average $R_{net,s}$ (at 10 m) during the daytime fog hours of all fogs in Table 2, excluding 28 Oct 2014 because the measurements are biased.

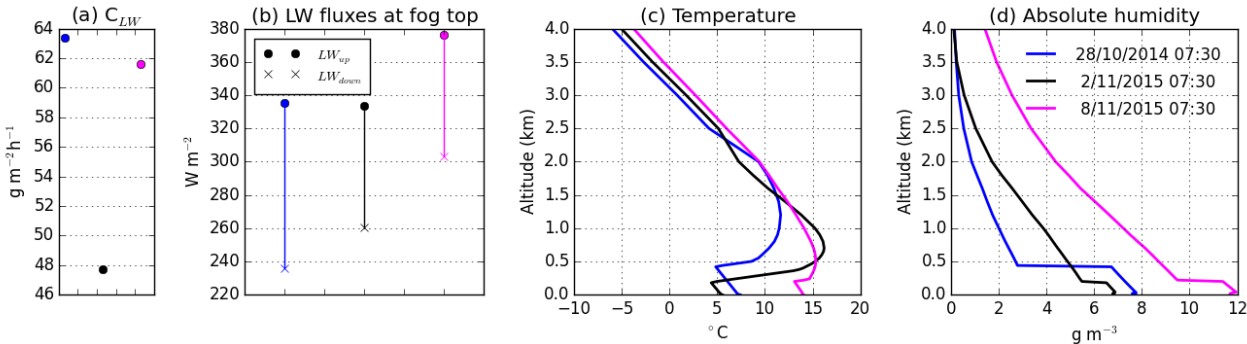

**Figure 6:** Comparison of three fog events at 07:30 UTC: (a) $C_{LW}$ (defined in Sect. 2.1); (b) LW fluxes at fog top (cross is downwelling, circle is upwelling, thus length of vertical line indicates the (negative) LW budget at fog top); (c) Temperature profile; (d) humidity profile. The fog top is located at the sharp drop in humidity.




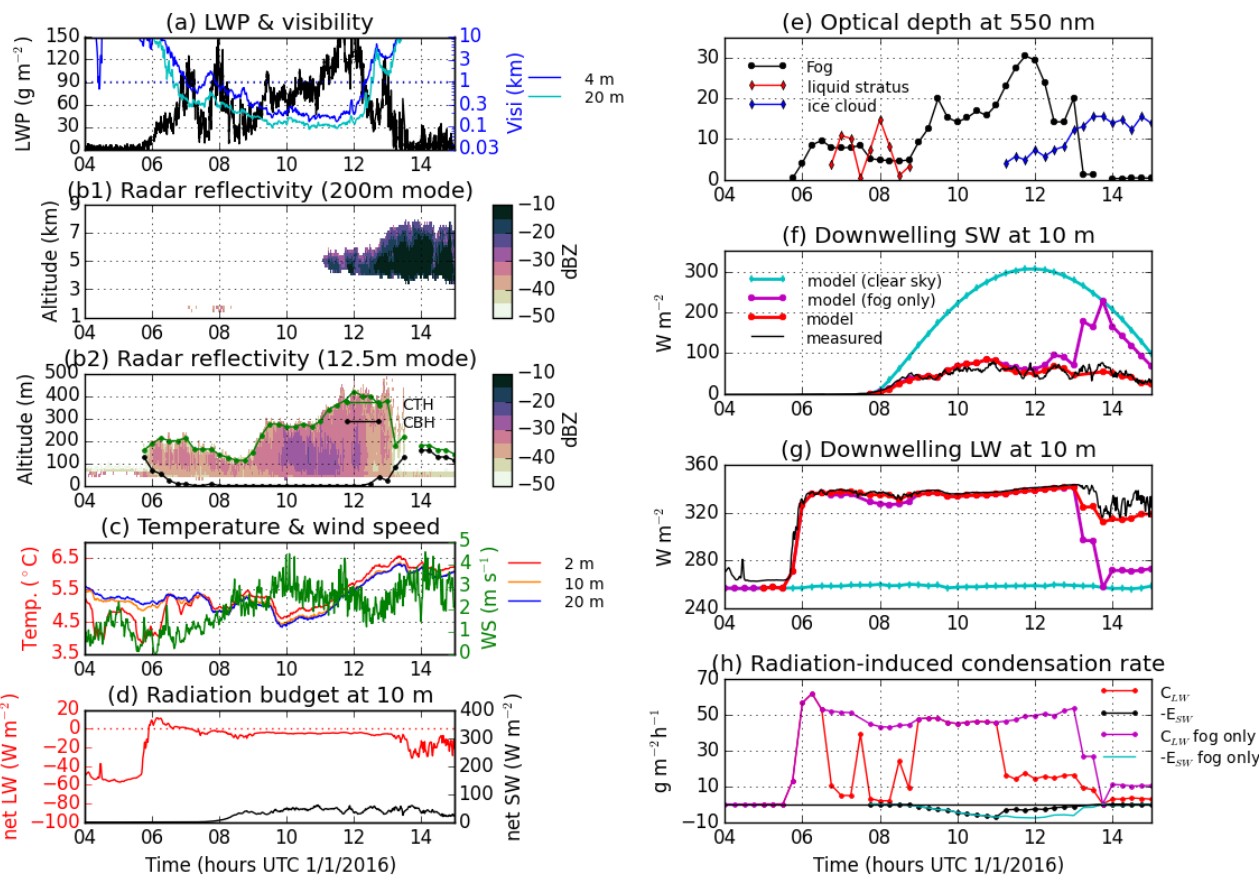

**Figure 7:** Case study of the fog event on 1 Jan 2016, when clouds appeared above the fog. Panels are the same as in Fig. 3, with a few additions: In (b), there are two panels, the upper one showing the reflectivity from the 200m mode of the radar and the lower one that of the 12.5m mode. In (e), the optical depths of the cloud layers above the fog are also indicated, and in (f–h) the results obtained when including only the fog (and not the higher clouds) have been added.





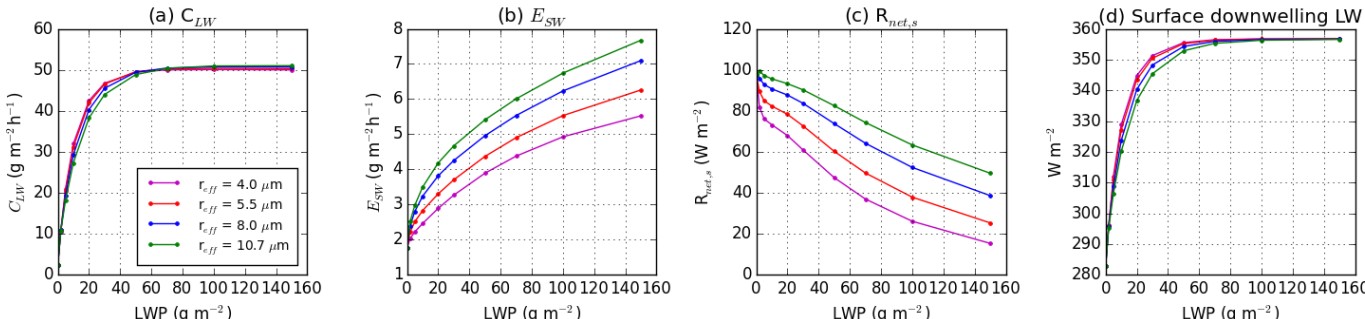

**Figure 8: Dependency of $C_{LW}$ (a), $E_{SW}$ (b), $R_{net,s}$ (c) (defined in Sect. 2.1), and the downwelling LW flux at the surface (d), on the fog LWP and effective radius. All other input data are fixed to the values of 27 Oct 2014 at 08:30 UTC: the fog is 100 m thick with no above clouds and a solar zenith angle of 73.9°.**

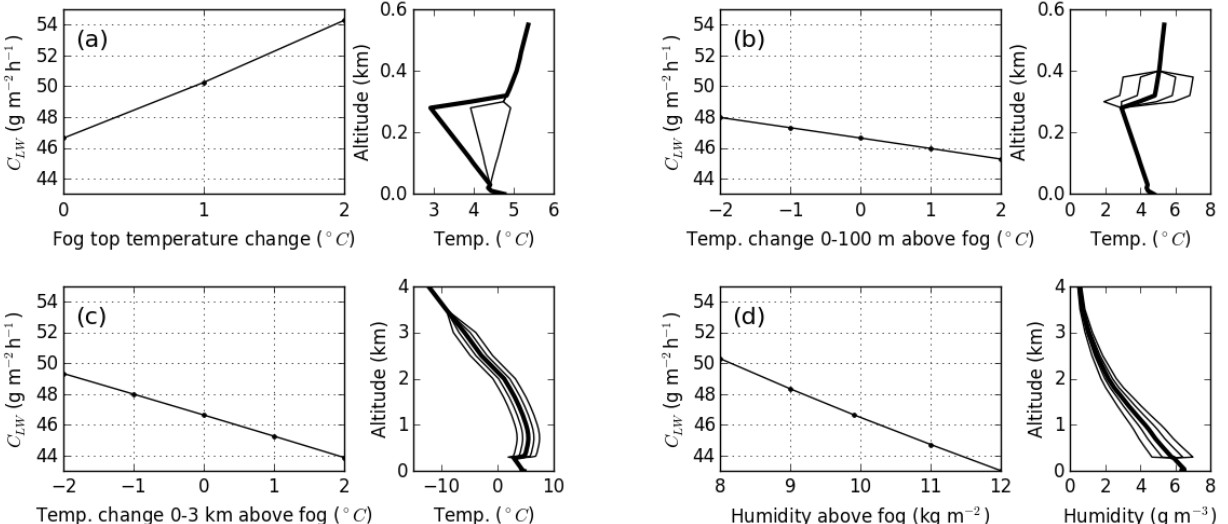

**Figure 9: Sensitivity of $C_{LW}$ (defined in Sect. 2.1) to changing the fog-top temperature (a), the temperature in the first 100 m above the fog (b), the temperature in the first 3 km above the fog (c) and the humidity above the fog (d). All other input data are kept constant at the values for 13 Dec 2015 at 10 UTC: the fog is 290 m thick with no clouds above and a visible optical depth of 16.4. To the right of each result is a plot showing how the profile of temperature or humidity is modified from the original profile (thick line).**




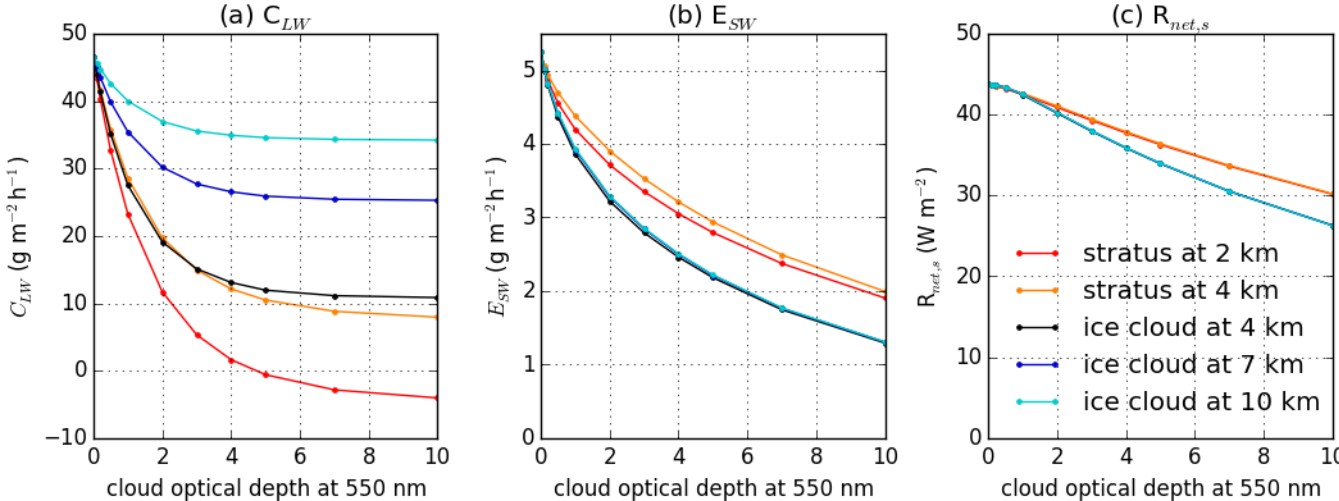

**Figure 10: Sensitivity of $C_{LW}$ (a), $E_{SW}$ (b) and $R_{net,s}$ (c) (defined in Sect. 2.1) to the altitude, type and visible optical depth of a cloud appearing above the fog. The tests are performed for the same situation as in Fig. 9. Solar zenith angle is 75.7°.**

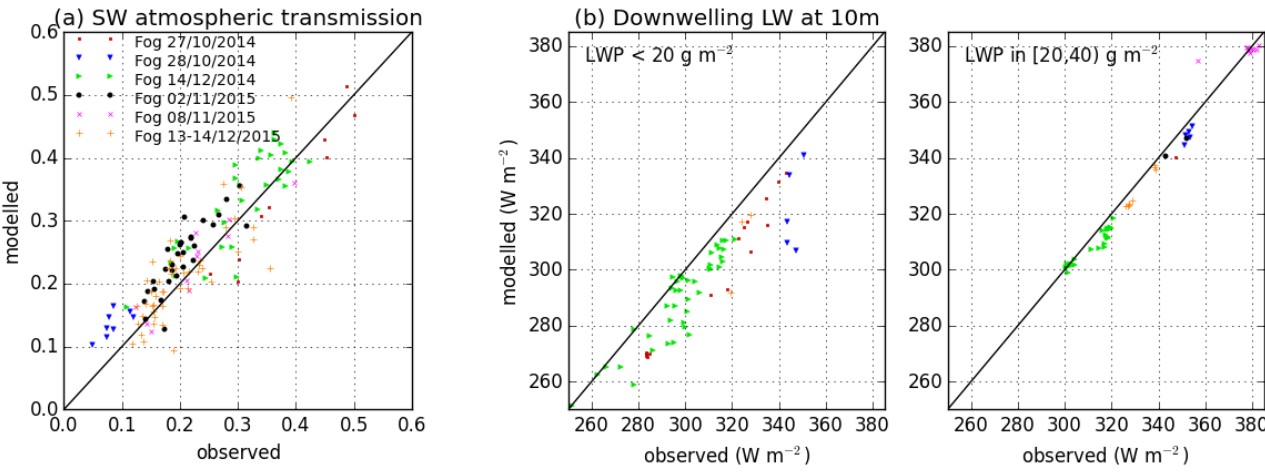

**Figure A1: Comparison of modelled and measured SW and LW downwelling radiative flux at 10 m during the six fog cases**
10 **without a higher cloud (Table 2): (a) Atmospheric SW transmission (fraction of downwelling SW at 10 m and at the top of the atmosphere), including only times when observed flux exceeds 10 W m⁻²; (b) downwelling LW flux at 10 m, in cases where fog LWP is estimated to less than 20 g m⁻² and between 20 and 40 g m⁻², respectively.**





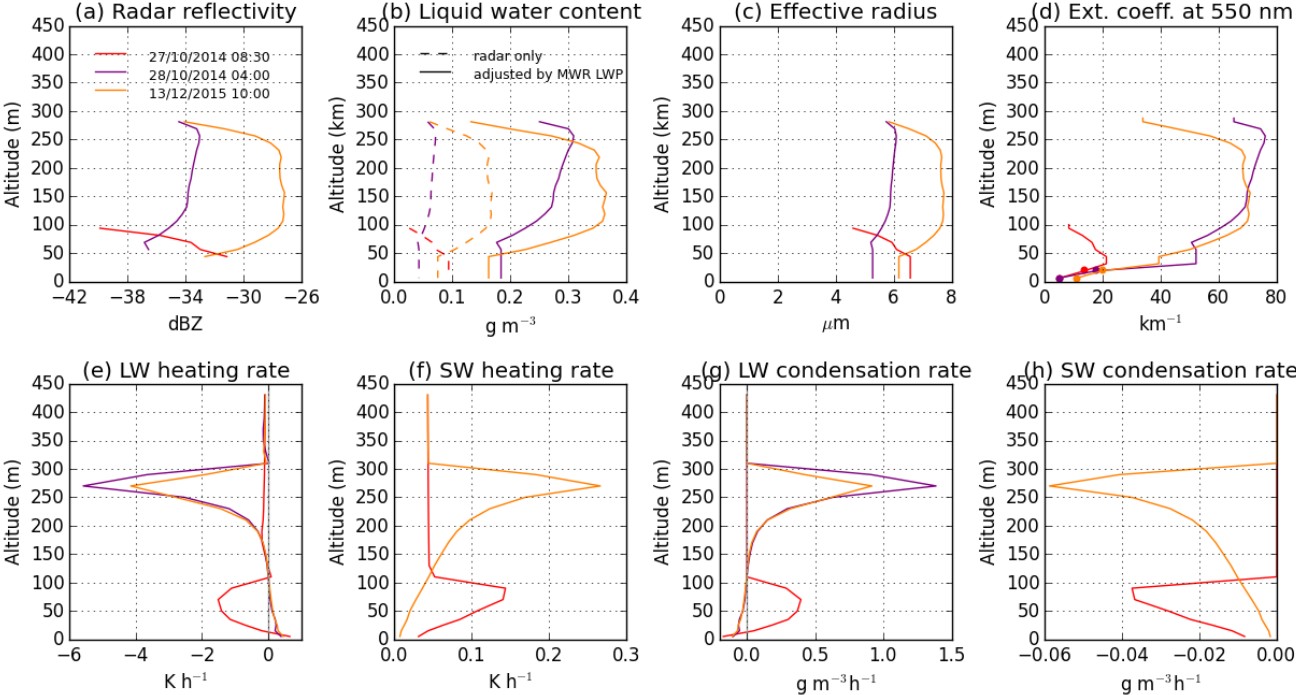

**Figure B1: For three different fog situations: Vertical profile of (a) 10min mean radar reflectivity, (b) LWC estimated with Eq. (8) before and after normalisation with the MWR LWP (normalisation not performed for red line, as LWP < 10 m$^{-2}$), (c) r$_{eff}$ estimated from Eq. (9), (d) visible extinction coefficient estimated from Eq. (6) (above 30 m) and from Eq. (10) (below 30 m; circles indicate estimates from the visibility meters), (e–f) radiative heating rate calculated from LW and SW radiation, and (g–h) the subsequently calculated condensation rates with Eq. (5). The solar zenith angle is similar in the two situations in day.**