# Peer review of "Radiation in fog: Quantification of the impact on fog liquid water based on ground-based remote sensing"

_Atmospheric Chemistry and Physics, 2017_

## Referee Comment (RC1) · Anonymous Referee #1 · 29 May 2017

The meticulous preparation of the authors has resulted in a manuscript that will pass most requirements for publication in ACP. Yet this referee is left with an uncomfortable feeling about this paper. Primary concern is the fact that the authors have omitted a thorough review and discussion of the kinetic energy budgets of the fog layers as a means to deal with radiative heating and cooling. Throughout the manuscript it is stressed that the LW radiation constitutes a source of LWC capable of renewing the entire fog in 1 – 2 hours (see f.e. pg 11, line 16; but there are several other places). Renewal means that there is a substantial sink of LWC. The only sink that this reviewer can think of is precipitation or 'wet deposition'. No credit or evidence is given to the existence of either of these two depletion mechanisms. If there is precipitation then

the radar signal would be swamped by it, but there is no evidence of that either in this paper. Consequently there should be at least some credit given to the possibility that the LW cooling at the top gives rise to downdrafts that will mix the fog layer and evaporate the air towards the bottom of the layer. In other words the LW cooling does not give rise to additional condensation but is a source of kinetic energy. Some of the profiles in the back (f.e. Fig 6) show T-profiles with large vertical gradients indicative of an adiabatic state that could potentially be the result of turbulent mixing due to LW-cooling. In addition LW-cooling converted to TKE at the top can drive entrainment of dry air from above the fog top into the fog layer. This is a conversion of potential energy into kinetic energy. It seems to me that this paper needs a convincing treatment of aspects of the TKE budget as it relates to LW cooling and heating in addition to the current treatise which only discusses LW cooling and condensation rates.

---

## Referee Comment (RC2) · Anonymous Referee #2 · 7 Jun 2017

**1   Main Review Points**

This paper presents a study of how fog liquid water is impacted by radiation balance. Ground-based remote sensing observations of seven fog events in the Paris region are used in the analysis. The manuscript is well-written, clearly structured and of a very high scientific quality. I highly recommend publication in ACP.

The main request I have is to somewhat more clearly highlight how representative or not the findings can be expected to be, in the main part of the manuscript as well as in the abstract.

[Figure]

**2 Minor Comments**

- Page 1, line 15 (henceforth 1-15 etc.): what is meant by 'renewing' here? Doubling? With a rate of 40gm-2h-1 or greater, it should be less than one hour, in this case.

- 1-17: 100% of what?

- 1-22: 30% of what?

- 7-2: Why 15 minutes? Is this the temporal scale at which you expect changes to occur?

- 7-2: Why only one event with clouds above, if you suspect this type of situation to be so important for radiation balance? What is the potential for generalization?

- 7-20: Why do you apply averaging to some parameters and not others?

- 7-26: Do you mean 'at the surface'? If not, how do you determine visibility for situations with cloud-base height 'close to the surface'?

- 7-26: Given that the ceilometer is mostly blind below $\sim 50\,\mathrm{m}$, what do you do with these situations, if they occur at all?

- 11-2: Please explain what is meant by 'renew' here (and in the following sections).

- 12-15: "The two parameters..." - Is this a qualitative statement? If not, can you provide a correlation coefficient, please?

- 13-5: Is there no way to test this (ice phase) assumption using measurements?

- Conclusions section: What can be learned from your findings to improve numerical weather prediction?

**3 Technical Remarks**

- 2-3: real time

- 2-5: Continental fog often forms by

- 2-22, 3-2: fogs → fog [I am not sure the plural exists, and there is no need to use it here.]

- 3-3: methodS or methodOLOGY

- 3-16: 10m above ground

- 3-24: wood → forest

- 7-10: fogs → fog [I am not sure the plural exists, and there is no need to use it here.]

- 8-4: only ON the liquid...

- 11-26: higher by 14g... on 8 Nov

- 14-27: above the fog ARE thus

- 18-11: rateS

- 21-34: is occurring → occurs

- 22-3: contributionS

- 30-Fig1: In my version of the manuscript, some of the text is somewhat compressed (Winiarek)

- 30-Fig3: Maybe decrease size of captions to avoid overlap with labels (as in b and h)

---

## Author Comment (AC1) · 23 Jul 2017

**Response to comments from Anonymous Referee #1**

**Referee comment:**

*The meticulous preparation of the authors has resulted in a manuscript that will pass most requirements for publication in ACP. Yet this referee is left with an uncomfortable feeling about this paper. Primary concern is the fact that the authors have omitted a thorough review and discussion of the kinetic energy budgets of the fog layers as a means to deal with radiative heating and cooling. Throughout the manuscript it is stressed that the LW radiation constitutes a source of LWC capable of renewing the entire fog in 1 – 2 hours (see f.e. pg 11, line 16; but there are several other places). Renewal means that there is a substantial sink of LWC. The only sink that this reviewer can think of is precipitation or 'wet deposition'. No credit or evidence is given to the existence of either of these two depletion mechanisms. If there is precipitation then the radar signal would be swamped by it, but there is no evidence of that either in this paper. Consequently there should be at least some credit given to the possibility that the LW cooling at the top gives rise to downdrafts that will mix the fog layer and evaporate the air towards the bottom of the layer. In other words the LW cooling does not give rise to additional condensation but is a source of kinetic energy. Some of the profiles in the back (f.e. Fig 6) show T-profiles with large vertical gradients indicative of an adiabatic state that could potentially be the result of turbulent mixing due to LW- cooling. In addition LW-cooling converted to TKE at the top can drive entrainment of dry air from above the fog top into the fog layer. This is a conversion of potential energy into kinetic energy. It seems to me that this paper needs a convincing treatment of aspects of the TKE budget as it relates to LW cooling and heating in addition to the current treatise which only discusses LW cooling and condensation rates.*

We would like to thank the referee for the instructive comments. From referee's text, we interpret the following 3 major comments:

**Comment 1:** The authors should consider the coupling between radiative cooling and vertical motions within the fog layer and its consequences for the condensation rates.

**Response 1:** We agree with the referee that the radiative cooling of the upper fog is closely related to a destabilisation and vertical mixing of the fog layer, at least when the fog is opaque to LW radiation so that the cooling occurs primarily near the fog top. This is why we assume that the cloud has an adiabatic temperature profile (see Sect. 3.3). When the fog is well-mixed, the radiative cooling at fog top will be rapidly spread uniformly in the fog so that the entire layer cools at approximately the same rate. Our current methodology calculates the condensation rates directly from the radiative cooling where it occurs. However, since the fog is assumed to be saturated everywhere, the vertically integrated condensation rate ($C_{LW}$) will not strongly depend on where in the fog the cooling occurs. The only change will be due to the lower parts of the fog being slightly warmer (up to 2 ˚C) than the fog top, so that the same cooling will give a larger condensation rate at the higher temperature (due to $\frac{\partial \rho_s}{\partial T}$ increasing with temperature, see Sect. 2.4). We have repeated the calculations of condensation rates with a vertically uniform cooling rate, and this causes

only 1–2 % increase in $C_{LW}$, which is insignificant. The relative effect on $E_{SW}$ is even smaller. In the revised manuscript, we have included a discussion of this point in Sect. 2.4.

We would like to stress that the temperature profiles shown in Fig. 6c are adiabatic in the fog because we prescribe this (see Sect. 3.3). Many radiosonde observations in fog support that (thick) fogs are adiabatic within and capped by a sharp inversion (see Sect. 3.3). In the revised manuscript, we have clarified this in the figure text.

**Comment 2:** The paper should explain better what are the sink processes of LWP which counteract the renewal of LWP by LW cooling.

**Response 2:** We agree with the referee that there are important sink processes of LWP in the fog, notably the entrainment of unsaturated air at fog top (which causes evaporation as it mixes with the fog) and evaporation of droplets near fog base (due to heating from the ground). The radar does not detect any rain during the fog cases studied in this paper, but a weaker deposition of cloud droplets at the surface might contribute to limiting fog LWP, as has been found in previous fog studies (e.g. Brown and Roach, 1976; Price et al., 2015). Although these sink processes are all affected by the radiative cooling through the vertical motions it generates, including them in our analysis would require a dynamical model in addition to the radiative transfer code. We therefore consider that the inclusion of these processes should be the topic of a separate publication. We would argue that it is possible to study the effect of radiation in fog without taking these sink processes into account, as long as this limitation is clearly explained (which we strive to do in the revised manuscript, Sect. 2.4). We realise that a more thorough discussion of the fog LWP sink processes in the paper is necessary to allow readers to understand how our results fit into the larger context of the fog life cycle. We have therefore added some more background on the sink processes in Sect. 1 and discussion of the link between our results and the sink processes in Sect. 5.2.

**Comment 3:** The paper needs a convincing treatment of the TKE budget of the fog, because the TKE budget is related to the radiative cooling.

**Response 3:** As already argued in response 2, we consider that the analysis of the turbulence of the fog layer and its relations to radiative cooling, fog-top entrainment and interactions between the fog and the surface, is outside the scope of this paper. In the introduction to the revised manuscript, we have added a description of the full system of fog processes and their interactions, including the role of TKE.

[revised manuscript text omitted]

---

## Author Comment (AC2) · 23 Jul 2017

**Response to comments from Anonymous Referee #2**

*This paper presents a study of how fog liquid water is impacted by radiation balance. Ground-based remote sensing observations of seven fog events in the Paris region are used in the analysis. The manuscript is well-written, clearly structured and of a very high scientific quality. I highly recommend publication in ACP.*

We would like to thank the referee for the thoroughly review of our manuscript and the many useful major and minor comments.

**1 Main review points:**

**Comment 1:** *The main request I have is to somewhat more clearly highlight how representative or not the findings can be expected to be, in the main part of the manuscript as well as in the abstract.*

**Response 1:** The representativeness of the findings is a very important question, and we agree with the referee that we should write more about it. From radiative transfer theory, we know that the main factors that determine variations in the divergence of radiative fluxes are temperature, humidity, clouds and aerosols. While the impact of aerosols is only preliminary studied in this paper, the temperature, IWV and fog LWP of the studied cases cover an important range of values (see Table 2 and Figure 5a), and the sensitivity study to higher clouds include both thin and thick clouds at several altitudes. Therefore, the variability found in the radiative processes includes an important range of conditions found in mid-latitude winter. However, the results cannot be directly applied to situations outside this range, such as very cold weather, when ice crystals may form in the fog, or in the tropics or in summer, when the screen temperature and IWV can be much higher. Even though this study only covers radiation fog and stratus-lowering fog occurring over a continental surface, our methodology to quantify the radiative processes should in principle be transferable to all fog types, since the radiation is not directly impacted by the formation mechanism. However, the parameters of the model could change if the method was applied to a different location; for example, a larger effective radius should be used in clean background areas with fewer cloud condensation nuclei, and a different surface albedo and emission temperature for fog above the sea. Fog events occurring in rain have been left out from this study, because the retrievals from both radar and MW radiometer are biased during rain. In the revised manuscript, we have included some of the considerations discussed above in the abstract and in the conclusions.

**2 Minor points**

**Comment 2.1:** *Page 1, line 15 (henceforth 1-15 etc.): what is meant by 'renewing' here? Doubling? With a rate of 40gm-2h-1 or greater, it should be less than one hour, in this case.*

**Response 2.1:** By 'renewing', we mean the time it takes before the LW cooling process have produced the same amount of liquid water as is currently in the fog. To understand the renewal time, one can look at Fig. 5a, where the LW-driven condensation is plotted against the current fog LWP. If the LW-induced condensation rate is fog LWP is 50 g m$^{-2}$ and the LW cooling produces 50 g m$^{-2}$ h$^{-1}$, the renewal takes 1 hour. It is correct that in some cases the renewal time is even shorter than 1 hour, such as the green dots near (LWP=25 g m$^{-2}$, C$_{LW}$=50 g m$^{-2}$) which have renewal time of 0.5 hours. To be more consistent with this, we have written "0.5–2 hours" instead of "1–2 hours" throughout the revised manuscript.

**Comment 2.2:** *1-17: 100% of what?*

**Response 2.2:** We mean that the LW radiative cooling rate of the fog is reduced by up to 100 % when a low cloud appears, relative to clear-sky conditions. We have written this more clearly in the revised manuscript.

**Comment 2.3:** *1-22: 30% of what?*

**Response 2.3:** The heating rate of the fog due to absorption of SW radiation inside the fog increases by 30 % when the aerosols are taken into account, relative to the heating rate when only liquid droplets are taken into account. We have written this more clearly in the revised manuscript.

**Comment 2.4:** *7-2: Why 15 minutes? Is this the temporal scale at which you expect changes to occur?*

**Response 2.4:** Yes, we expect significant changes to occur on timescales longer than 15 minutes, which we can also see from the time series of e.g. LWP and cloud thickness (see for example Fig. 7). We also want to average out the high-frequent variability in LWP, which is due to instrumental noise or small-scale heterogeneity of the fog. Although small-scale fog heterogeneity is found to be of importance for fog dissipation (e.g. Bergot, 2016), it is not suitable to study this with a radiation model only, as this requires a treatment of the dynamics of the fog (e.g. by Large-Eddy simulations). With the methodology of our paper, a quantification of the mean impact of the radiation on the fog as a whole is more suitable. The exact choice of 15 minutes between each calculation is rather arbitrary.

**Comment 2.5:** *7-2: Why only one event with clouds above, if you suspect this type of situation to be so important for radiation balance? What is the potential for generalization?*

**Response 2.5:** The fog observation dataset at SIRTA did not include many cases with a non-precipitating cloud above the fog which was detected by the radar. In precipitating conditions, the retrieval of cloud optical properties only from radar is challenging because the cloud cannot easily be separated from the trailing rain below (this exclusion of rainy cases is explicitly mentioned in section 2.5 of the revised manuscript). Due to this lack of suitable cases where higher clouds occurred, we chose to focus mainly on the sensitivity study for the impact of higher clouds (Sect. 5.2). The results of this sensitivity study should be applicable to most clouds, since we study the effect on both altitude and cloud optical

depth, which are the two main factors determining the cloud radiative effect (e.g. Dupont and Haeffelin, 2008).

The methodology for estimating the radiative properties of the higher clouds occurring on 1 January 2016 was specific for these clouds. To generalize, we recommend to derive the cloud emission temperature from the altitude of the cloud (as seen by the radar) and then separate the clouds in two classes, optically thin (for which LW emissivity is less than 1) and optically thick (emissivity 1), inferred from thickness and the radar reflectivity. Such a methodology could be tuned or validated using simultaneous satellite observations of the clouds from which the LW emissivity and SW albedo can be derived. In the revised manuscript, we mention this perspective in the conclusions.

**Comment 2.6:** *7-20: Why do you apply averaging to some parameters and not others?*

**Response 2.6:** The radar, ceilometer, visibility and MWR LWP observations are given at high temporal resolution (1 minute or less) and have high-frequent variability. In order to calculate radiation representative for a longer time window, we therefore average these quantities in time. The MWR temperature and humidity profiles are only given every 5 minutes and have less fast variability than the other observations; it therefore seemed reasonable to pick the closest profile in time instead of averaging two profiles. The radiosondes are only launched twice a day, so time averaging is not feasible.

**Comment 2.7:** *7-26: Do you mean 'at the surface'? If not, how do you determine visibility for situations with cloud-base height 'close to the surface'?*

**Response 2.7:** The wording "close to the surface" was used because we also check if the cloud-base is below 20 m by using the visibility measured at 20 m. Since the ceilometer has a vertical resolution limited to 15 m, the cloud-base height can in some cases be adjusted up or down to 20 m by using the information about the visibility here. However, as this is a minor detail, we agree that it is better to write "The cloud-base height is set to 0 m if visibility at 4 m is below 1 km". This is done in the revised manuscript.

**Comment 2.8:** *7-26: Given that the ceilometer is mostly blind below ~50 m, what do you do with these situations, if they occur at all?*

**Response 2.8:** The CL31 ceilometer deployed at SIRTA does not have a blind zone. We therefore can detect cloud-base height with the ceilometer as low as 7.5 m, although it is true that several effects introduce uncertainty in the first gates, such as the partial overlap of transmitter and receiver. However, for cloud-base detection, the precision in the backscatter signal is not very critical. We also use the visibility meter at 20 m to verify if the cloud base is above or below 20 m, using the 1 km horizontal visibility threshold. The CL31 is described in the paper by Kotthaus et al. (2016); we have included this reference in the revised paper.

**Comment 2.9:** *11-2: Please explain what is meant by 'renew' here (and in the following sections).*

**Response 2.9:** We refer to the answer to comment 2.1. In the revised manuscript, we have also added some explanation of what we mean by renewal at the first place we use it (page 11, line 2 in the old manuscript version).

**Comment 2.10:** *12-15: "The two parameters..." - Is this a qualitative statement? If not, can you provide a correlation coefficient, please?*

**Response 2.10:** We have calculated the correlation coefficient (R=0.56) and included it in the revised manuscript.

**Comment 2.11:** *13-5: Is there no way to test this (ice phase) assumption using measurements?*

**Response 2.11:** Among the ground-based remote sensing instruments for cloud observation at SIRTA, only the backscatter lidars with dual polarization are able to distinguish between ice and liquid particles. These instruments cannot see through the fog, so we therefore cannot verify if the cloud has liquid or not. However, as we already mention in the paper, the MWR LWP goes to zero after the fog cloud dissipates and the higher cloud still is there, which is an indication that the cloud is purely ice.

**Comment 2.12:** *Conclusions section: What can be learned from your findings to improve numerical weather prediction?*

**Response 2.12:** The important effect on the fog LWP budget caused by clouds appearing above the fog could be studied in the context of numerical weather prediction (NWP). Based on the results of this paper, an NWP study of fog with and without elevated clouds could be designed to study how the presence of multilayer clouds affects NWP of fog life cycle. We have added this perspective in our Conclusions section.

**3 Technical remarks**

We thank the referee for having spotted all these small technical issues.

**Comments on English language and typing errors:**
• *2-3: real time*
• *2-5: Continental fog often forms by*
• *2-22, 3-2: fogs → fog [I am not sure the plural exists, and there is no need to use it here.]*
• *3-3: methodS or methodOLOGY*
• *3-16: 10m above ground*
• *3-24: wood → forest*
• *7-10: fogs → fog [I am not sure the plural exists, and there is no need to use it here.]*
• *8-4: only ON the liquid...*
• *11-26: higher by 14g... on 8 Nov*
• *14-27: above the fog ARE thus*
• *18-11: rateS*
• *21-34: is occurring → occurs*

*• 22-3: contributionS*

**Response:** The language errors mentioned by the referee have all been fixed in the revised manuscript. We have replaced the plural "fogs" with "fog", "fog events" or "fog situations" wherever it occurred. We also spotted that an exponent 2 was missing for "T" in Clausius–Clapeyron's equation on page 6, line 10, and we have corrected this.

**Comments on figure layout:**
*• 30-Fig1: In my version of the manuscript, some of the text is somewhat compressed (Winiarek)*
*• 30-Fig3: Maybe decrease size of captions to avoid overlap with labels (as in b and h)*

**Response:** We have reduced the title font size in Figs. 3,4,7 to avoid the overlap and fixed the problem with the text in Fig. 1 in the revised manuscript.

[revised manuscript text omitted]